# Scale Alone Does not Improve Mechanistic Interpretability in Vision Models

**Roland S. Zimmermann**[1]*      **Thomas Klein**[1,2]*      **Wieland Brendel**[1]

## Abstract

In light of the recent widespread adoption of AI systems, understanding the internal information processing of neural networks has become increasingly critical. Most recently, machine vision has seen remarkable progress by scaling neural networks to unprecedented levels in dataset and model size. We here ask whether this extraordinary increase in scale also positively impacts the field of mechanistic interpretability. In other words, has our understanding of the inner workings of scaled neural networks improved as well? We use a psychophysical paradigm to quantify one form of mechanistic interpretability for a diverse suite of nine models and find no scaling effect for interpretability — neither for model nor dataset size. Specifically, none of the investigated state-of-the-art models are easier to interpret than the GoogLeNet model from almost a decade ago. Latest-generation vision models appear even less interpretable than older architectures, hinting at a regression rather than improvement, with modern models sacrificing interpretability for accuracy. These results highlight the need for models explicitly designed to be mechanistically interpretable and the need for more helpful interpretability methods to increase our understanding of networks at an atomic level. We release a dataset containing more than $130'000$ human responses from our psychophysical evaluation of 767 units across nine models. This dataset facilitates research on automated instead of human-based interpretability evaluations, which can ultimately be leveraged to directly optimize the mechanistic interpretability of models.

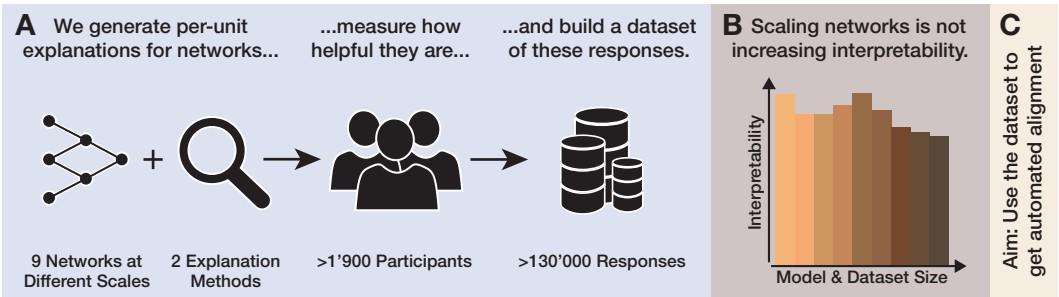

Figure 1: **Has scaling models in terms of their dataset and model size improved interpretability? A.** We perform a large-scale psychophysics experiment to investigate the interpretability of nine networks through the two most-used mechanistic interpretability methods. **B.** We see that scaling has not led to increased interpretability. Therefore, we argue that one has to explicitly optimize models to be interpretable. **C.** We expect our dataset to enable building automated measures for quantifying the interpretability of models and, thus, bootstrap the development of more interpretable models.

*Equal contribution. [1] Max Planck Institute for Intelligent Systems, Tübingen AI Center, Tübingen, Germany [2] University of Tübingen, Tübingen AI Center, Tübingen, Germany. Correspondence to: research@rzimmermann.com. Code & Dataset: brendel-group.github.io/imi.

37th Conference on Neural Information Processing Systems (NeurIPS 2023).

# 1 Introduction

Since the early days of deep learning, artificial neural networks have been referred to as black boxes: opaque systems that learn complex functions which cannot be understood, not even by the people who build and train them. Mechanistic interpretability [37] is an emerging branch of explainable AI (XAI) focused on understanding the internal information processing of deep neural networks, possibly by focusing on individual units as their atomic building blocks. This line of research is akin in spirit to the early days of neuroscience, where the receptive fields of cells in the mammalian visual cortex were investigated using single-cell electrophysiology [22]. Designing interpretable neural networks and aligning their information processing with that of humans would not only satisfy academic curiosity but also constitute a major step toward trustworthy AI that can be employed in high-stakes scenarios.

A natural starting point for mechanistic interpretability research is to investigate the individual units of a neural network. For convolutional neural networks (CNNs), the individual output channels of a layer, called activation maps, are often treated as separate units [38]. A common hypothesis is that channel activations correspond to the presence of features of the input [38]. There is hope that by understanding which feature(s) a unit is sensitive to, one could build a fine-grained understanding of a model by identifying complex circuits within the network [5]. To learn about a unit's sensitivity, researchers typically focus on inputs that cause strong activations at the target unit, either by obtaining highly activating images from the training set (*natural exemplars*), or by generating synthetic images that highly activate the unit. The well-known method of feature visualization [12, 38] achieves this through gradient ascent in input space (see Sec. 3.2). However, in practice, identifying a unit's sensitivity is far from trivial [4]. Historically, work on feature visualization has focused on the Inception architecture [47], in particular GoogLeNet. But in principle, both of these methods should work on arbitrary network architectures and models.

The starting hypothesis of this work is that the dramatic increase in both the scale of the datasets and the size of models [7, 45] might benefit per-unit mechanistic interpretability. Evidence for this hypothesis comes from recent work showing that models trained on larger datasets become more similar in their decisions to human judgments as measured by error consistency [14]. It is conceivable that models make more human-like decisions because they rely on non-spurious/human-aligned features. Therefore, one can argue that networks with more human-like decisions are more interpretable. Another argument for the hypothesis that scale is beneficial for unit-wise interpretability is that as models get larger, they can dedicate more units to represent learned features without having to encode features in superposition [10]. This could render the units more interpretable since the image features that activate them become less ambiguous.

We conduct a large-scale psychophysical study (see Fig. 1) to investigate the effects of scale and other design choices and find *no practically relevant* differences between any of the investigated models. While scaling models and datasets has fuelled the progress made on many research frontiers [7, 19, 25], it does not improve the mechanistic interpretability of individual units. Neither scale nor the other design choices make individual units more interpretable on their own.

As our study shows, new model design choices or training objectives are needed to *explicitly* improve the mechanistic interpretability of vision models. We expect the data collected in our study to serve as a starting point and test bed to develop cheap automated interpretability measures that do not require collecting human responses. These automated measures could pave the way for new ways to directly optimize model interpretability. Therefore, we release the study's results as a new dataset, called *ImageNet Mechanistic Interpretability* (IMI), to foster new developments in this line of research.

# 2 Related Work

The idea of investigating the information processing on the level of individual units in neural networks has a long history [e.g., 2, 53, 3, 32], possibly inspired by work in the neuroscience community that investigates receptive fields of individual neurons [e.g., 1, 39], dating back as far as the seminal work of Hubel and Wiesel [22] which categorized cells in the cat's visual cortex into simple and complex cells. The same holds for the technique of feature visualization, first proposed by Erhan et al. [12], developed further by, e.g., Mahendran and Vedaldi [31], Nguyen et al. [35], Mordvintsev et al. [33], Yosinski et al. [51], and popularized by Olah et al. [38]. Ghiasi et al. [16] present work

on extending feature visualizations to ViTs. Nguyen et al. [36] experimented with imposing priors on feature visualizations to make them more similar to natural images. Kalibhat et al. [24] aim to improve the interpretability afforded by natural exemplars by finding natural language descriptions of units through CLIP models [40]. Only years after the work on improving feature visualizations matured was their usefulness for understanding units experimentally quantified by Borowski et al. [4] and Zimmermann et al. [54], who found that feature visualizations are helpful but not more so than highly activating natural exemplars. Recently, Geirhos et al. [15] demonstrated that feature visualizations are not guaranteed to be reliable and might be misleading.

Much work on interpretability has focused on so-called post-hoc explanations, that is, explaining specific model decisions to end users [e.g. 41, 46, 26]. In contrast, mechanistic interpretability [37], the branch of XAI that we focus on here, is concerned with understanding the internal information processing of a model. This approach is not limited to the interpretability of single features we investigate here but also encompasses the analysis of entire circuits [5] and investigations of phase changes that occur over the course of training [34], to name just a few examples. See the review by Gilpin et al. [17] for a distinction and a broader overview of the field of XAI.

As Leavitt and Morcos [28] point out, it is vitally important to not only generate explanations that look convincing but also to conduct falsifiable hypothesis testing in interpretability research, which is what we attempt here. Furthermore, as Kim et al. [27] emphasize, interpretability should be evaluated in a human-centric way, a stance that motivates employing a psychophysical experiment with humans in the loop to measure interpretability. The field of interpretability has always struggled with a lack of consensus about definitions and suitable measurement scales [8, 29, 6]. Several previous works [e.g. 44, 20, 50, 27] focus on measuring the utility of post-hoc explanations. In contrast, we here are not primarily concerned with methods that explain model decisions to end-users, but instead focus on introspective methods that shed light on the internal information processing of neural networks.

Our psychophysical experiment builds on work by Borowski et al. [4] and Zimmermann et al. [54], whose psychophysical task we expand and adapt for arbitrary models as outlined in Sec. 3.2.

## 3 Methods

### 3.1 Measuring the Mechanistic Interpretability of Many Models

**Selecting Models.** We investigate nine computer vision models compatible with ImageNet classification [42]. These models span four different design axes, allowing us to analyze the influence of an increasing model scale on their interpretability. First, we look at the influence of model size in terms of parameter count, starting with GoogLeNet [47] at 6.8 million parameters and culminating in ConvNeXt-B [30] at 89 million parameters. Next, we look at various model design choices, such as increasing the width or depth of models (GoogLeNet vs. ResNet-50 [18] vs. WideResNet-50 [52] vs. DenseNet-201 [21]) and using different computational blocks (ViT-B [9] vs. ConvNeXt). Third, we scale training datasets up and compare the influence of training on 1 million ImageNet samples to pre-training on 400 million LAION [45] samples (ResNet-50 vs. Clip ResNet-50 [23, 40] and ViT-B vs. Clip ViT-B [23]). Last, we test the relation between adversarial robustness and interpretability (ResNet-50 vs. Robust ResNet-50 [43, 48]) as previous work [11, 49] found adversarial robustness to be beneficial for feature visualizations.

**Selecting Units.** For each of the investigated models, we randomly select 84 units (see Appx. A.5) by first drawing a network layer from a uniform distribution over the layers of interest and then selecting a unit, again at random, from the chosen layer. This scheme is used instead of randomly drawing units from a uniform distribution over all units since CNNs typically have more units in later layers. The layers of interest are convolution and normalization layers, as well as the outputs of skip connection blocks. We avoid the very first convolution layers since they can be interpreted more directly by inspecting their filters [38, 4]. For GoogLeNet, we select only from the last layers of each inception block in line with earlier work [4, 54]. For the ViT models, we adhere to the insights by Ghiasi et al. [16] and only inspect the position-wise feedforward layers.

**Performing & Designing the Psychophysics Experiment.** As interpretability is a human-centric model attribute, we perform a large-scale psychophysical experiment to measure the interpretability of models and individual units. For this, we use the experimental paradigm proposed by Borowski

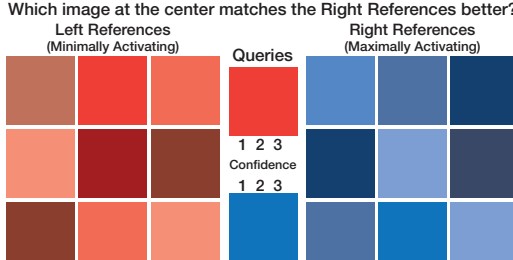

Which image at the center matches the Right References better?

Left References
(Minimally Activating)

Queries

Right References
(Maximally Activating)

1 2 3
Confidence
1 2 3

Figure 2: **Illustration of task design.** Users see a set of nine maximally/minimally activating reference images (synthetic feature visualizations or natural exemplars) on the right/left side of the screen. In the center, one strongly positively and a strongly negatively activating natural image are shown. Users need to pick the more positively activating query image (here, the bottom one) by pressing on a number indicating their confidence in their choice. See Fig. 8 for an example.

et al. [4] and Zimmermann et al. [54]: Here, the ability of humans to predict the sensitivity of units is used to measure interpretability. Specifically, crowd workers on Amazon Mechanical Turk complete a series of 2-Alternative-Forced-Choice (2-AFC) tasks (see Fig. 2 for an illustration). In each task, they are presented with a pair of strongly and weakly activating (query) images for a specific unit and are asked to identify the strongly activating one. During this task, they are supported by 18 explanatory (reference) images that strongly activate the unit, either natural dataset exemplars or synthetic feature visualizations. We begin by making the task as easy as possible by choosing the query images as the most/least activating samples from the ImageNet dataset. By choosing query images that cause less extreme activations, the task's difficulty can be increased, which allows us to probe a more general understanding of the unit's behavior by participants. For details refer to Appx. A.1.

While we explain the task to the participants, we do not instruct them to use specific strategies to make their decisions to avoid biasing results. For example, we do not explicitly prompt them to pay attention to the colors or shapes in the images. Instead, participants complete at least five hand-picked practice trials to learn the task and receive feedback in all trials. Once they have successfully solved the practice trials, they are admitted to the main experiment, in which they see 40 real trials interspersed with five fairly obvious catch-trials. See Appx. A.2 for details on how trials are created. In all trials, subjects give a binary response and rate their confidence in their decisions on a three-point Likert scale. For each investigated model, we recruit at least 63 unique participants who complete trials for 84 randomly selected units of each model (see Appx. A.5). This means every unit is seen by 30 different participants. Within each task, no unit is shown more than once. We ascertain high data quality through two measures: First, by restricting the worker pool to experienced and reliable workers. Second, by performing quality checks and excluding participants who show signs of not paying attention, such as failing to get all practice trials correct by the second attempt, failing to pass catch trials, taking too long, or being unreasonably quick. We also forbid workers to participate multiple times in our experiments to avoid biases introduced through learning effects. We keep recruiting new participants until 63 workers pass our quality checks per model. See Appx. A.3 for details.

We finally refer to the ratio of correct answers as *interpretability score* and use it as a measure of a unit's interpretability. As there are two options participants have to choose from, random guessing amounts to a baseline performance of 0.5. We record $> 130'000$ responses from $> 1'900$ unique participants recruited over Amazon Mechanical Turk for 767 units spread across 9 models. For more details, refer to Appx. A.1.

### 3.2 Scaling Feature Visualization to Many Models

Feature visualization describes the process of synthesizing maximally activating images through gradient ascent on a unit's activation. While simple in principle, this process was refined to produce the best-looking visualizations (see Sec. 2). However, these algorithmic design choices and the required hyperparameters have predominantly been optimized for a single model — the original GoogLeNet. This poses a challenge when creating synthetic feature visualizations for different models, as required for a large-scale comparison of models such as ours: How should these hyperparameters be chosen for each model individually without introducing any biases to the comparison? While we cannot revisit all algorithmic choices, we develop an optimization procedure for setting the most crucial parameters, i.e., the number of optimization steps and the strength of the regularizer responsible for creating visually diverse images. In a nutshell, we stop optimization based on the achieved relative activation value and perform a binary search over the latter hyperparameter, to obtain feature visualizations that

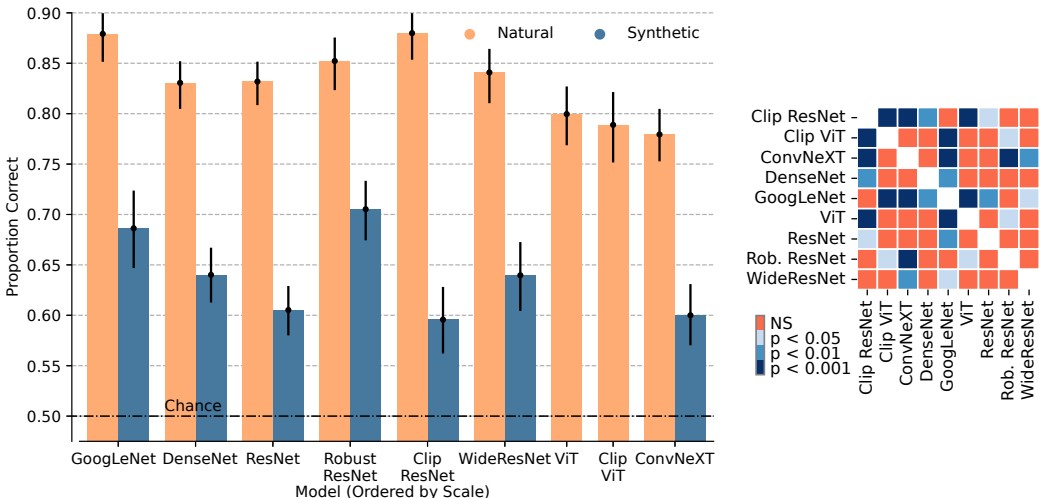

Figure 3: **Left. Model size and training schemes have little influence on per-unit mechanistic interpretability.** We compare the mechanistic interpretability of the units of nine vision models for two interpretability methods: maximally activating dataset samples (Natural) and feature visualizations (Synthetic). In a large-scale psychophysical experiment, we compare models that differ in architecture, training objectives, and training data. While these models reflect the advancements in model design in recent years (sorted by model size first and then dataset size), we surprisingly see little to no effect of these design choices on mechanistic, per-unit interpretability. While these results might appear promising as all models yield scores of about $80\%$ (natural), note that we demonstrate that interpretability is far more limited than it first appears and breaks down dramatically as the task is made harder in Sec. 4.4. Also, note that error bars represent confidence intervals around the estimated means, not variance of the underlying data (see also Sec. 4.5). **Right. Few models have significantly different interpretability scores.** The differences across models in interpretability afforded by natural exemplars are mostly non-significant (NS) in a Conover test with Holm correction for multiple comparisons; see Fig. 11 for significance values for synthetic feature visualizations.

are comparable in terms of how well they activate a unit. For details, see Appx. A.4. Unfortunately, there is no generally accepted method for generating feature visualizations for ViT models yet: While Ghiasi et al. [16] present a method to generate visualizations for ViTs, we refrain from using it because one of the steps of their procedure seems hard to justify (see Appx. A.4).

## 4 Results

We now present and analyze the data we obtained through our psychophysical experiment. We look at how scaling models affects mechanistic interpretability (Sec. 4.1), compare feature visualizations and exemplars (Sec. 4.2), investigate systematic layer-dependence of interpretability (Sec. 4.3), and investigate the dependence of our results on task difficulty (Sec. 4.4). Lastly, we introduce a dataset bundling the experimental data that we hope can lead to new avenues for mechanistic interpretability research (Sec. 4.5). Unless noted otherwise, error bars correspond to the 95th percentile confidence intervals of the mean of the unit average estimated through bootstrap sampling.

### 4.1 Scaling Models Does not Coincide with Improving Interpretability

We begin by visualizing the interpretability of the nine networks investigated in Fig. 3 for both the natural and the synthetic conditions. We sample models with different levels of scale (in terms of model or dataset size) and different training paradigms, but find little to no difference in their interpretability. Strikingly, the latest generation of vision models (i.e., ConvNeXT and ViT) performs *worse* than even the oldest model in this comparison (GoogLeNet).

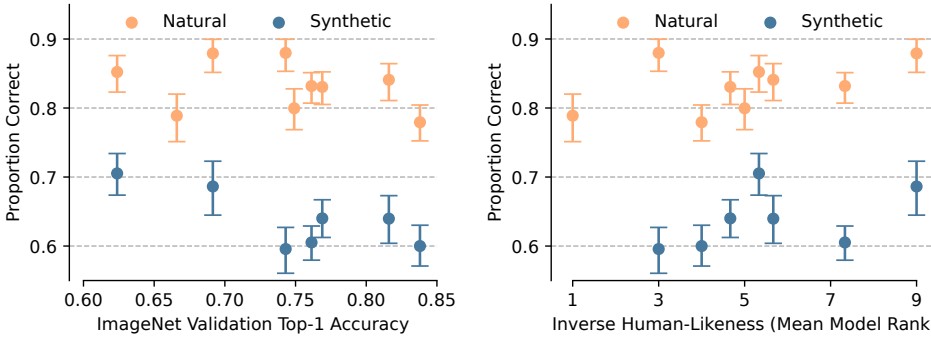

Figure 4: **Neither higher classification performance nor more human-like decisions come with higher interpretability. Left.** While the investigated models have strongly varying classification performance, as measured by the ImageNet validation accuracy, their interpretability shows less variation for both natural exemplars (orange) and synthetic feature visualizations (blue). More accurate classifiers are not necessarily more interpretable. For synthetic feature visualizations, there might even be a regression of interpretability with increasing accuracy. **Right.** A similar result is obtained when quantifying the similarity models have to human behavior. This similarity is measured by the mean rank statistic of the model-vs-human benchmark [14], with a lower rank meaning that the model is more human-like.

We similarly see no improvements if we plot a model's interpretability as a function of how similar it behaves to humans. For this, we use two metrics: For one, the model's classification performance on ImageNet, for another, a measure of consistency between a model's and human decisions [14]. In Fig. 4, we investigate the relationship between these two similarity measures and a unit's intepretability for both feature visualizations and natural exemplars. While models vary widely in terms of their classification performance ($\sim 60\,\%$ to $\sim 85\,\%$), their interpretability varies in a much narrower range for each method (see Fig. 4a (Left)). For feature visualizations, we see a decline in interpretability as a function of classification performance. For natural exemplars, we do not find any dependency between interpretability and classification performance. We find analogous results for the other similarity metric (see Fig. 4b (Right)). These results highlight that mechanistic interpretability, of the kind investigated here, does not directly benefit from scaling effects, neither in model nor dataset size.

## 4.2   Feature Visualizations are Less Helpful than Exemplars for all Models

The data in Fig. 3 clearly shows that the findings by [4] generalize to models other than GoogLeNet: Feature visualizations do not explain unit activations better than natural exemplars, regardless of the underlying model. This includes adversarially robust models, which have previously been argued to increase the quality of feature visualizations [11, 49]. The idea was that for non-robust models, naive gradient ascent in pixel space leads to adversarial patterns. To overcome this problem, various image transformations, e.g., random jitter and rotations, are applied to the image over the course of feature visualization. As adversarially more robust models have less adversarial directions, one can hope to obtain visualizations that are visually more coherent and less noisy. There is indeed a substantial and significant increase in performance in the synthetic condition for the robust ResNet-50 over the normal ResNet-50. In fact, this model significantly outperforms all models except GoogLeNet (see Fig. 11). Nevertheless, it remains true that natural exemplars are still far more helpful. To see whether well-interpretable units for one interpretability method are also well-interpretable for the other, we visualize them jointly in Fig. 12. Here, we find a moderate correlation between the two for a few models but no general trend.

## 4.3   Which Layers are More Interpretable?

In light of the small differences between models regarding the average per-unit interpretability, we now zoom in and ask whether there are rules to identify well-interpretable units within a model.

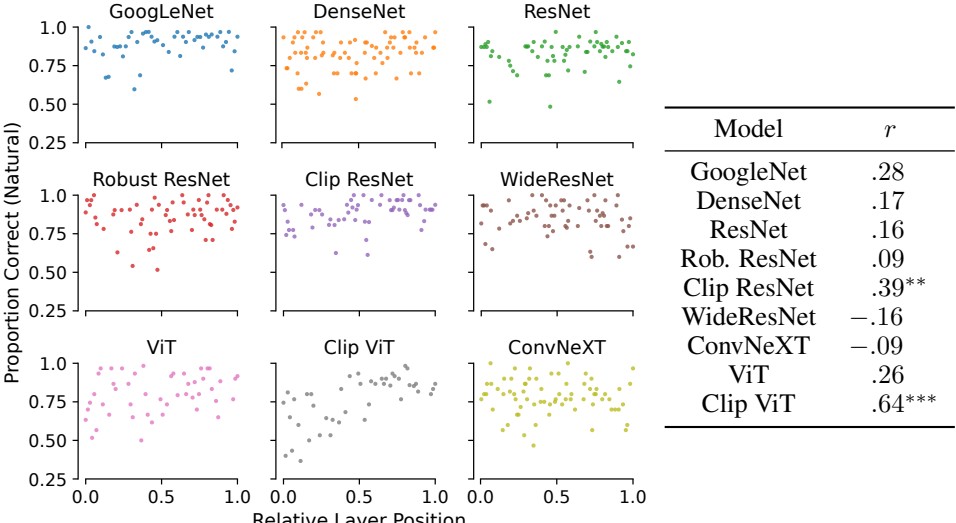

Figure 5: **The position of a layer is sometimes predictive of its interpretability.** We investigate the interpretability afforded by natural exemplars as measured in our psychophysical experiment by visualizing it for different units of various layers for all investigated networks as a function of their relative position within the network. Here, the first layer corresponds to a relative position of $0$, whereas the last layer has a position of $1$. The table shows Spearman's rank correlation between the proportion correct (averaged over multiple units from the same layer) and the layer position. Asterisks denote significant correlations using the thresholds shown in Fig. 3b (Right).

A unit's interpretability is not well predicted by its layer's position relative to the network depth (i.e., early vs. late layers). In Fig. 5, we visualize the recorded interpretability scores for all investigated layers as a function of their relative position.[2] We average the interpretability over all investigated units from a layer to obtain a single score per layer. To check for correlations between layer position and interpretability, we compute Spearman's rank correlation for the data of each model. For most models, we do not see a substantial correlation. However, two notable outliers exist: the Clip ResNet and Clip ViT. A strong and highly significant correlation can be found for both of them. We find much smaller correlations for the same architectures trained on smaller datasets (i.e., ResNet and ViT, trained on ImageNet-2012). We thus conclude that (pre-)training on large-scale datasets might benefit the interpretability of later layers while sacrificing that of early layers.

## 4.4 Do our Findings Depend on the Difficulty of the Task?

As outlined in Sec. 3.1, the difficulty of the task used to quantify interpretability depends on how the query images (i.e., the images that participants need to identify as the more/less strongly activating image) are sampled. So far, we have made the task as easy as possible: The query images were chosen as the most/least strongly activating samples from the entire ImageNet dataset. In this easy scenario, the models were all substantially more interpretable than a random black box (for which we would expect a proportion correct of $0.5$). We now ask: Are these models still interpretable in a (slightly) stronger sense, or do their decisions become incomprehensible to humans when increasing the task's difficulty ever so slightly? For this, we repeat our experiment for two models (ResNet-50 and Clip ResNet-50) with query images that are now sampled from the 99th (medium difficulty), 95th (hard difficulty) or 85th (very hard difficulty) percentile of the unit's activations. As the interpretability scores for synthetic feature visualizations are already fairly low in the previously tested easy condition (see Fig. 3a (Left)), we do not test them in the hard condition. Note that the reference images serving as explanations are always chosen from the very end of the distribution of activations, i.e., they are the same for all three difficulties.

---

[2]Note that the layer position is not precisely defined for layers computed in parallel, e.g., in the Inception blocks of the GoogLeNet architecture.

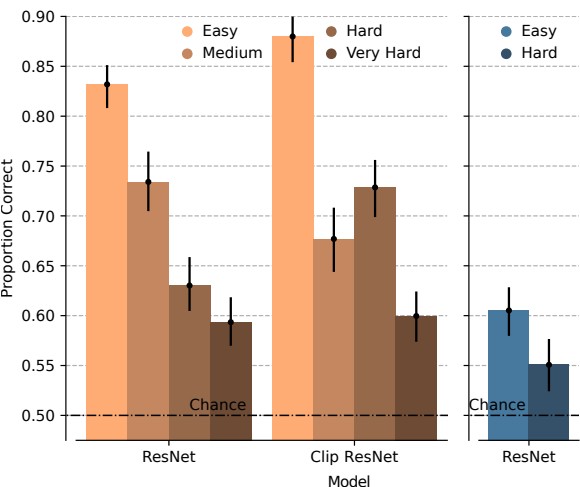

Figure 6: **Human performance decreases with increasing task difficulty.** We increase the task difficulty by not using the most strongly/weakly activating images as the query images (easy) but instead sampling them from the 99th (medium), 95th (hard) or 85th (very hard) percentile. We see a decrease in human performance with increasing difficulty. Strikingly, even a small change in the sampling (easy vs. medium) leads to stark performance decreases when using natural exemplars (left), showing that human understanding of a unit's overall behavior is relatively limited. For the synthetic feature visualizations, the performance is reduced close to chance level by this small change (right).

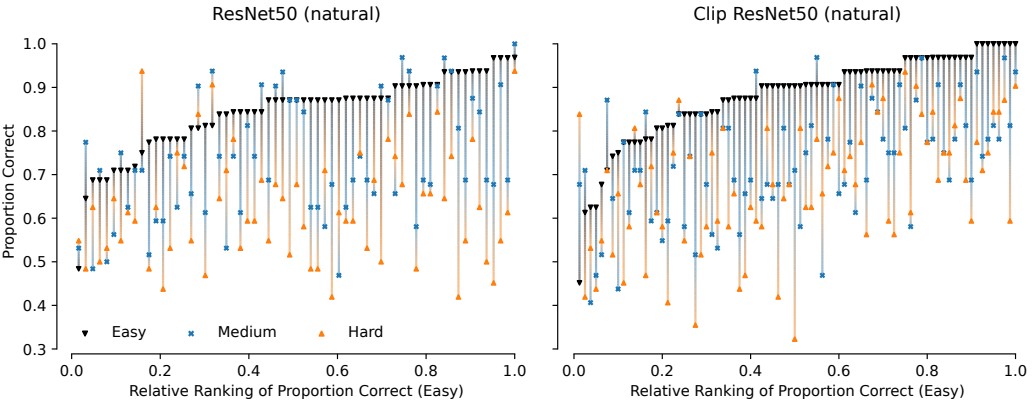

Figure 7: **Well-interpretable units do not necessarily stay interpretable in harder tasks.** We visualize the human performance for each unit investigated of the (Clip) ResNet-50 for the easy (black), medium (blue), and hard (orange) tasks in the natural condition. The units are ordered by the recorded proportion correct values in the easy task. As expected, the performance for almost all units decreases with increasing hardness. However, how much the performance drops is not strongly correlated with performance in the easy task, i.e., well-interpretable units in the easy condition do not necessarily stay well-interpretable in the harder task. For an alternative visualization that displays the gap between the difficulty levels separately, see Fig. 10.

The results in Fig. 6 show a drastic drop in performance when making the task only slightly more difficult (medium). For the synthetic feature visualizations, performance is reduced close to chance level. When looking at how the performance changes per unit (see Fig. 7), we see that for almost all units, the measured interpretability scores do indeed follow the defined difficulty levels, meaning that humans perform best in the easy and worst in the hard task.

But is this a fair modification of the task or does it make the task unreasonably difficult? If the distribution of activations for a unit across the entire dataset was multimodal with small but pronounced peaks at the end for strongly activating images and if we assume each of these modes corresponds to different behavior, making the task harder as described above would be unfair: When the query images are sampled from the 95th percentile while the reference images are still sampled from the distribution's tail, these two sets of images could come from different modes, which might correspond to different types of behavior, making the task posed to participants less meaningful. However, we find a unimodal distribution of activations that smoothly tapers out (see Fig. 16). In other words, the query images used in the harder conditions are in the same mode of unit activation as the ones from the easy condition, and we would, therefore, expect them to also be in a similar behavioural regime.

### 4.5 IMI - A Dataset to Learn Automated Interpretability Measures

The results above paint a rather disappointing picture of the state of mechanistic interpretability of computer vision models: Just by scaling up models and datasets, we do not get increased interpretability for free, suggesting that if we want this property, we need to *explicitly* optimize for it. One hurdle for research in this direction is that experiments are costly due to the requirement of human psychophysical evaluations. While those can be afforded for some units of a few models (as done in this work), it is infeasible to evaluate an entire model or even multiple models fully. However, this might be required for developing new models that are more interpretable. For example, applying the experimental paradigm used in this work to each of the roughly seven thousand units in GoogLeNet would amount to obtaining more than 200 thousand responses costing around 25 thousand USD. One conceivable way around this limitation is to remove the need for human evaluations by developing *automated* interpretability evaluations aligned with *human* judgments. Put differently, if one had access to a model that can estimate the interpretability of a unit (as perceived by humans), we could potentially leverage this model to directly optimize for more interpretable models.

To enable research on such automated evaluations, we release our experimental results as a new dataset called *ImageNet Mechanistic Interpretability* (IMI). Note that this is the *first* dataset containing interpretability measurements obtained through psychophysical experiments for multiple explanation methods and models. The dataset contains $> 130'000$ anonymized human responses, each consisting of the final choice, a confidence score, and a reaction time. Out of these $> 130'000$ responses, $76'000$ passed all our quality assertions while the rest failed (some of) them.[3] We consider the former to be the main dataset and provide the latter as data for development/debugging purposes. Furthermore, the dataset contains the used query images as well as the generated explanations for 767 units across nine models.

The dataset itself should be seen as a collection of labels and meta information without fixed features that should be predictive of a unit's interpretability. While there seem to be no large differences between models, there are considerable differences between individual units, even within the same model (e.g., see Fig. 5). Finding and constructing features that are predictive of these differences will be one of the open challenges posed by this line of research. We illustrate how this dataset could be used by trying to predict a unit's interpretability from the pattern of its activations in Appx. B.4 in two examples: First, we test the hypothesis that easier units are characterized by a clearly localized peak of activation within the activation map, while for harder units, the activation is more distributed, making it harder for humans to detect the unit's sensitivity. However, we do not find a reliable relationship between measures for the centrality of activations, e.g. the local contrast of activation maps, and the unit's interpretability. Second, we analyze whether more sparsely activated units, i.e., units sensitive to a very particular image feature, are easier to interpret as the unit's driving feature might be easier to detect and understand by humans. Similar to the other hypothesis, we also do not find a meaningful relation between the sparseness of activations and a unit's interpretability.

We deliberately do not suggest a fixed cross-validation split: Depending on the intended use case of models fit on the data, different aspects must be considered resulting in other splits. For example, when building a metric that has to generalize to different models, another split might be used than when building a measure meant to work for a single model only. For that reason, we recommend researchers to follow best practices when training models on our dataset.

## 5    Discussion & Conclusion

**Discussion**    Due to the costly nature of psychophysical experiments involving humans, we cannot test every vision model but had to make a selection. To perform the most meaningful comparisons and obtain as informative results as possible, we chose the four design axes outlined above and models representing different points along each axis. For some axes, we did not test all conceivable models, such as the largest vision model presented so far [7] as the weights have not been released yet. However, based on the trends in the current results, it is unlikely that the picture would drastically change when considering more models.

---

[3]Of the $57'310$ rejected responses, $10'570$ were only rejected because they came from crowd workers who participated more than once; see also Appx. A.3.

An explicit assumption of the approach to mechanistic interpretability investigated here is that feature representations are axis-aligned, i.e., features are encoded as the activations of individual units instead of being encoded using a population code. This can be motivated by the fact that human participants do not fail in our experiments completely — they achieve better than chance-level performance. Therefore, this approach of investigating a network does not seem to be entirely misguided, but that alone does not exclude other coding schemes.[4] Furthermore, Fig. 12 reveals that the two interpretability methods we investigated here are only partially correlated, so other explanation methods might come to different conclusions.

Assessing the interpretability of neural networks remains an ongoing field of research, with no clear gold standard yet. This work utilizes an established experimental paradigm to quantify human understanding of individual units within a neural network. While it is possible that the construction of a new paradigm may alter the results, we contend that the employed experimental paradigm closely mirrors how mechanistic interpretability is applied in practice. Additionally, one could argue that the models analyzed in this work are already interpretable — we just have not discovered the most effective explanation method yet. Although this is theoretically possible, it is important to note that we employed the two best and most widely-used explanation methods currently available, and we were unable to detect any increase in interpretability when scaling models up. We encourage further research on interpretability methods.

**Conclusion**  In this paper, we set out to answer the question: Does scale improve the mechanistic interpretability of vision models at the level of individual units? By running extensive psychophysical experiments and comparing various models, we conclude that none of the investigated axes seem to positively affect model interpretability: Neither the size of the model nor that of the dataset nor model architecture or training scheme improve interpretability. This result highlights the importance of building more interpretable models: Unless we explicitly design models with interpretability in mind, we do not get it for free by just increasing downstream task performance. We believe that the benchmark dataset we released can play an important enabling role in this line of research.

## Author Contributions

RSZ and WB conceived the idea for the project as a continuation of their earlier work, TK joined at an early stage. RSZ lead the project. WB initiated and supervised the project. RSZ and TK jointly implemented and conducted the experiment, building heavily on the existing setup by RSZ, with advice and feedback from WB. TK contributed code to extend the preparation of natural and synthetic stimuli to support multiple models with help from RSZ. The a priori power analysis was done by TK. RSZ conducted the final analysis and was responsible for the figures with contributions from TK. The manuscript was written jointly by RSZ and TK with advice from WB.

## Acknowledgements

We thank Evgenia Rusak, Felix Wichmann, Matthias Kümmerer, Matthias Tangemann, Robert Geirhos and Robert-Jan Bruintjes for their valuable feedback (in alphabetical order) and Max Wolff for his explorative research. This work was supported by the German Federal Ministry of Education and Research (BMBF): Tübingen AI Center, FKZ: 01IS18039A. WB acknowledges financial support via an Emmy Noether Grant funded by the German Research Foundation (DFG) under grant no. BR 6382/1-1 and via the Open Philantropy Foundation funded by the Good Ventures Foundation. WB is a member of the Machine Learning Cluster of Excellence, EXC number 2064/1 – Project number 390727645. This research utilized compute resources at the Tübingen Machine Learning Cloud, DFG FKZ INST 37/1057-1 FUGG. The authors thank the International Max Planck Research School for Intelligent Systems (IMPRS-IS) for supporting RSZ and TK.

---

[4]See work by Elhage et al. [10] for further arguments.

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

# A    Methodological Details

## A.1    Measuring the Mechanistic Interpretability of Many Models

To measure the interpretability afforded by a model, we extend the paradigm established by [4]. Participants in our study complete a sequence of 2-Alternative-Forced-Choice (2-AFC) trials, where each trial measures the interpretability of one unit of a network. In each trial, participants are presented with two so-called *query images*, sourced from the training set of ImageNet. One query image is highly positively activating for the investigated unit, i.e., feeding this image through the network would cause a large positive activation at the target unit. In contrast, the other query image is highly negatively activating. Participants are tasked with determining which of the two query images is the positive one. To do so, they are presented with two sets of nine *reference images* which characterize the unit. One set contains highly positively activating images, while the other contains highly negatively activating images. In the *natural* condition, these reference images are other natural images, whereas in the *synthetic* condition, the reference images are synthetic images generated by Feature Visualization. See Fig. 8 for an example of one trial in the natural condition. We phrase the task by asking which set of reference images fits the positive query image better so that participants can be completely agnostic with respect to the true semantics of the task. We also do not give overly specific instructions to avoid biasing the participants' behavior. Instead, participants learn the task by completing at least five hand-picked practice trials at the beginning of the experiment. Participants give a binary response and rate their confidence in their decision on a three-point Likert scale.

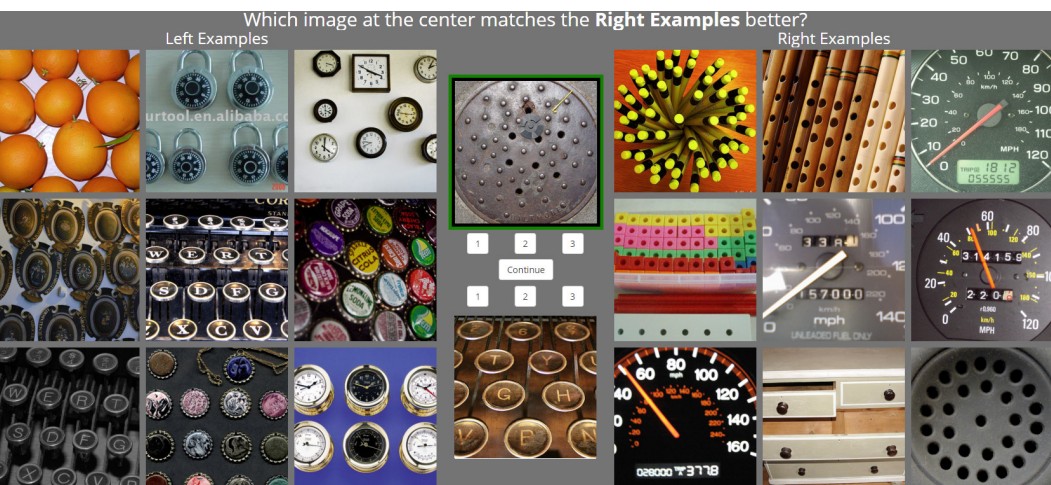

Figure 8: **Example of one trial.** What a crowd worker sees after having completed one trial: Two query images in the middle, two blocks of nine reference images to the sides, instructions, and feedback in the form of the green frame around the correct query image. Of course, this feedback is shown only after a correct response. In case of an incorrect response, the frame would be red.

## A.2    Sampling Images for the Psychophysical Tasks

The difficulty of an individual trial depends to a certain degree on the specific images that are shown in the trial. To avoid biasing the results for an individual unit, we do not only select the single highest/lowest activating image as a query image but instead create $t$ different trials for each unit. For each of these, we collect responses from crowd workers thrice. In the following, we describe the stimuli selection process for positively activating images, with negatively activating images being selected analogously. This procedure is similar to that of Borowski et al. [4], who also illustrate the approach in more detail. First, we select the top $9 \cdot t$ activating images as candidates for reference images, where $t$ is the number of unique trials to be generated. Then, we select the next $t$ images to be used as query images. To ensure that the range of activations yielded by the reference images does not differ across the $t$ tasks, we use the following procedure: We divide the range of candidate images into 9 groups of $t$ images each and create a set of reference images by sampling one image from each of the 9 groups without replacement. We initially create $t = 20$ trials but use only 10 of those, keeping the rest for an anticipated later experiment.

### A.3 Amazon Mechanical Turk

Our psychophysical study is conducted on Amazon Mechanical Turk to meet the requirement of scale. To maintain high data quality, we exclude participants who do not fulfill certain criteria. First of all, we restrict participation in our experiment to countries in which workers can be expected to be adequately proficient in English and in which completion of our click-work at the expected hourly wage is not unreasonably more profitable than other work, which we deemed unethical. Specifically, we restrict participation to the USA, Canada, Great Britain, Australia, New Zealand, and Ireland. As a second barrier, we only offer our Human Intelligence Task (HIT) to experienced workers who have submitted at least $2'000$ HITs for which the response was approved. To ascertain high reliability, we further restrict the pool to workers whose approval rate is at least 99%. Of course, we also prevent workers from participating in our experiments more than once[5]. Even if workers meet the aforementioned requirements, they might still be distracted during the experiment or give random answers to quickly finish the experiment (e.g., if they are unmotivated or frustrated due to the task difficulty). Therefore, we filter our data further. To use only data from workers who understand the task, we only accept HITs that require no more than three attempts at solving the demo trials and reject workers who spend less than 15 seconds reading the instructions. To catch workers who click mindlessly, we exclude responses in which fewer than four of our five catch-trials were answered correctly and responses that take the worker less than 135 seconds overall. On the other hand, we also reject responses that take them longer than $2'500$ seconds since it can be assumed that these workers interrupted their work. We also reject responses in which participants select the same query image (as in, the upper / lower one) in more than 90% of trials.

We recruit participants for each investigated model and experimental condition until 63 unique participants pass our quality checks. The responses of the workers who have not passed these checks are not used in our analysis but are included in our IMI dataset. Each participant completes at least 5 practice trials to get used to the task, 40 real trials, and 5 catch trials with obvious, hand-picked stimuli. In total and excluding pilot experiments, we collect data for $133'310$ trials, of which $76'000$ pass all quality checks.

We select 84 units of each model so that every unit is seen by 30 different participants since, within each task, no unit is shown more than once. All procedures conform to Standard 8 of the American Psychological 405 Association's "Ethical Principles of Psychologists and Code of Conduct" (2016). Participants are compensated at a targeted hourly rate of 15 USD, which amounts to 2.79 USD per task.

### A.4 Scaling Feature Visualization to Many Models

A fundamental problem with using natural images to characterize the receptive field of individual units (apart from idiosyncrasies of the used dataset) is that visual features do not usually appear in isolation, resulting in ambiguity. For example, highly activating ImageNet-exemplars for a unit sensitive to feathers would probably depict birds, making it hard to isolate feathers as the crucial visual feature instead of beaks, claws, or a background of greenery or blue sky.

The promise of Feature Visualization is to circumvent these limitations by synthetically generating images that only contain visual features contributing to high unit activation. The procedure starts with an initial random noise image and performs gradient ascent on the activation achieved by this image at the unit of interest. Following established work [e.g. 4, 54], a unit is defined as one feature map of a convolutional layer, where the activation across the feature map is aggregated by calculating the mean, just like for natural stimuli. To prevent mode collapse of the generated batch of feature visualizations, i.e. to truthfully capture the receptive field of so-called polysemantic units that show sensitivity to multiple different concepts, a regularization term is added to the loss to diversify the images.

We build on an existing implementation [38] and extend it to support various models flexibly. Previous implementations had two critical hyperparameters: the number of gradient ascent steps to be performed and the weight used for the diversity term. As earlier work mainly focused on the GoogLeNet model, hyperparameters were tuned for it. We find, however, that these fixed values do

---

[5]Due to technical issues, some workers participated more than once. However, we exclude their data in our analysis and recollect the missing data by recruiting new participants.

not generalize well to other models, but their optimal[6] values heavily depend, among other factors, on the model and location of the unit within the network — in extreme cases, the ideal value can even be different for two units of the same layer in the same network. Therefore, using any fixed value would introduce an unfair bias for or against some models. Furthermore, since a larger weight for the diversity term hinders the optimization, the number of necessary gradient ascent steps depends partially on the diversity weight, meaning these parameters cannot be set independently.

To overcome the latter problem of choosing an appropriate number of optimization steps, we implement an adaptive procedure that interrupts the optimization when the gradients become small. The procedure performs at least $2'500$ steps of gradient ascent and records a trajectory of the observed gradient magnitude. We smooth these trajectories with a large sliding window and halt optimization once the average gradient magnitude in the last window is larger than in the second-to-last window.

To solve the first problem, we determine the diversity weight for each unit individually as follows. We first record the maximum and minimum activation achieved by natural dataset samples for the unit. Then, we generate feature visualizations without diversity and assert that they achieved a stronger activation. We then try to find the largest possible diversity value that still produces images that achieve at least as strong activations as all dataset samples. To do so, we first perform an exponential search starting at a diversity of $1$, increasing by a factor of $10$ in each step. Once the value becomes too large, we perform 6 steps of binary search between the largest diversity value still known to work and the final value tested in the exponential search. If no value tested during the binary search worked, we return the lower bound of the search range, i.e. the images generated in the end are always guaranteed to be at least as activating as the strongest natural images. Generating one batch of Feature Visualizations, i.e., one step of the procedure, takes between two and $90$ minutes on an Nvidia 2080Ti GPU, depending mostly on the width of the layer of the unit, since the diversity term scales quadratically. A qualitative comparison of feature visualizations generated for the different models considered in this work can be found in Fig. 9.

For ViTs, feature visualization could theoretically be performed using the same method by maximizing the activation at the position-wise feedforward layers. However, just applying the existing methodology does not lead to visually coherent images. Ghiasi et al. [16] present a method for adapting the procedure to ViTs that seems to produce intelligible images, but one step of their algorithm just adds large-scale noise to the visualizations, effectively performing a random search in image space to find activating images. Removing this augmentation or reducing the scale of the noise leads to unintelligible images again. In light of these issues, we chose not to evaluate ViTs in the synthetic condition.

---

[6]Judged by the first authors.

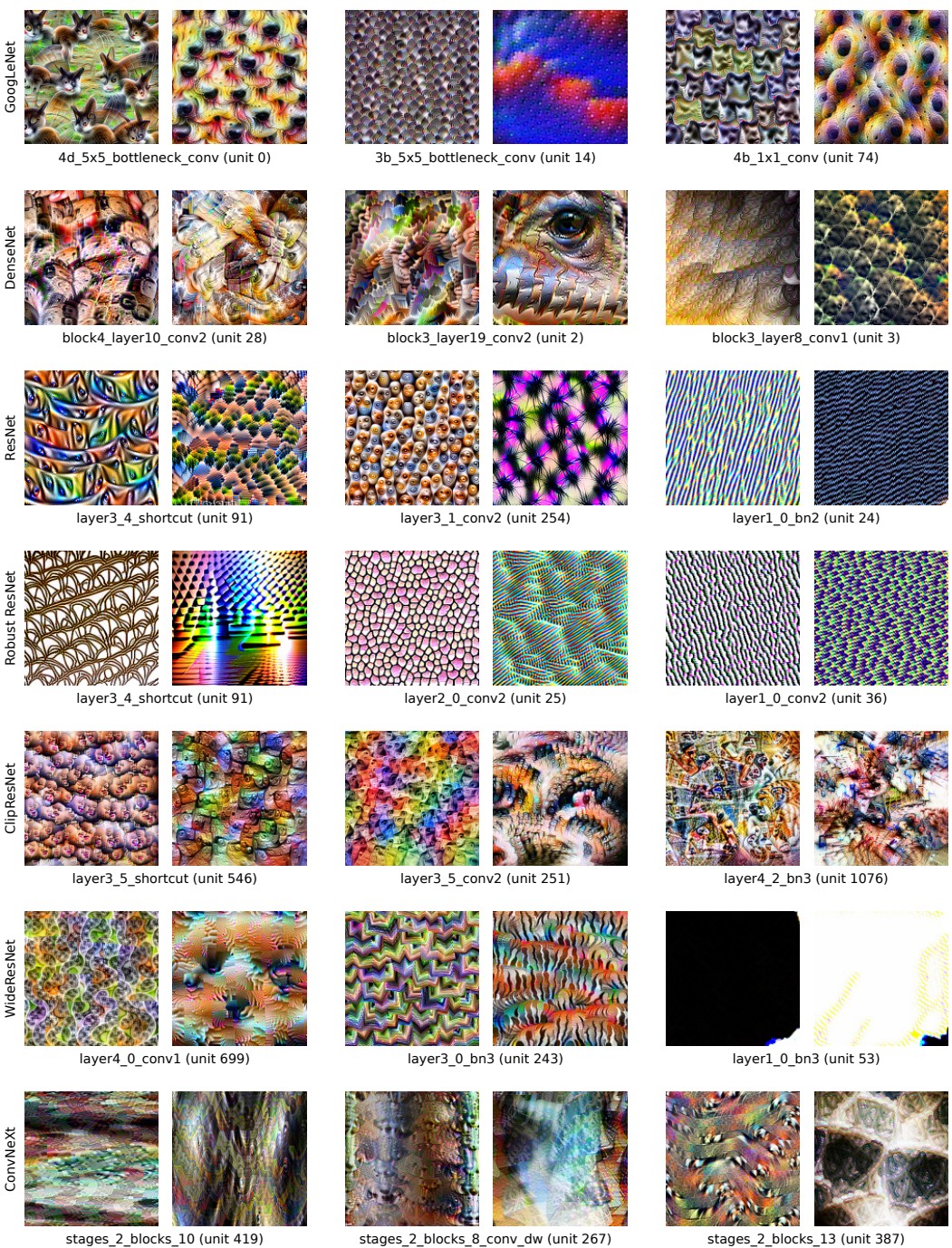

Figure 9: **Qualitative Comparison of Feature Visualizations.** For each model, we randomly choose three units and display the maximally (left) and minimally (right) activating feature visualizations generated without the diversity regularizer.

## A.5 A Priori Power Analysis

A central question for the experimental design of this study is how many units need to be sampled per model to obtain a result representative of the entire model. Answering this question is non-trivial as there might be large inter-unit interpretability differences within one model. Indeed, this is what we observe as displayed in Fig. 5). While the most naive approach would be to test all units, this is unfeasible due to the associated financial costs. Therefore, we need to find a trade-off between these considerations and keep the number of sampled units as low as possible while still getting representative results. Put differently: What is the lowest number of units one can select while still being reasonably sure that the found effect is statistically significant?

To answer this question, we first ran a pilot study where we controlled for inter-participant differences by showing stimuli from two models (GoogLeNet and Robust ResNet-50) to the same subjects. Participants in this pilot were the study's first authors and other lab members. This means that the obtained data is of high quality, and we can be confident that all participants understood the task. The mean difference in the proportion of correctly completed trials came out to be $0.1$, with standard deviations of $0.15$ for both interpretability methods, resulting in a relatively large effect size, with Cohen's $d$ of $0.67$. Irrespective of concerns of statistical significance, we deem an effect of this size to be practically relevant; in other words, if the difference in interpretability between two models would be at least $10$ percentage points, we would consider this practically relevant. To determine the required number of sampled units at these effect sizes, we then performed an a-priori power analysis using the software G*Power [13] — a standard tool widely used in psychology and the social sciences. To avoid unrealistic assumptions about the shape of the distribution of measurements (the normality-assumption of the t-test will almost certainly not be met because the data points are proportions expected to lie between $0.5$ and $1.0$), we opted for the non-parametric Mann-Whitney-U test. We assumed an $\alpha$-level of $0.01$ (subject to Bonferroni-correction to safely conduct up to five significance tests on the same data) and a $\beta$-level of $0.95$. This analysis yields that at least $86$ units are required.

However, the situation is further complicated by the fact that we are comparing values of which we cannot actually take a continuous measurement since we aggregate binary trials to estimate the proportion of correctly completed trials for each unit, i.e. there is measurement noise. This can be modeled as a Binomial distribution, characterized by the parameter $p$, the probability of answering correctly in any given trial for units of this model. This gives rise to the question of how many measurements we should take per unit to be able to assess an individual unit's interpretability with any confidence. Accepting a standard deviation of $0.1$ in the estimate of each unit's $p$ results in $30$ independent trials per unit.

Another consideration is how many trials one participant can be asked to complete. Earlier work presented up to $24$ trials to each participant under similar conditions [54]. Still, again we might be interested in accurately estimating the participant's performance, and each participant incurs some fixed cost for the time spent instructing them and completing the practice trials. On the other hand, MTurk HITs are typically very short. Constructing long tasks, e.g. of $100$ trials or more, would increase the risk of participants losing focus or becoming frustrated and just answering randomly. We deemed $55$ trials per participant ($40$ real trials, $10$ instruction trials, and $5$ catch trials) a suitable balance of these concerns.

Finally, the required number of participants is the total number of trials divided by the number of trials per participant. The total number of trials is, of course, the number of units times the number of necessary measurements per unit, resulting in $86 \cdot 30/40$ trials. As this is not an integer, we opt for using $84$ units instead, which brings the number of needed participants to $63$.

# B Further Experimental Results

## B.1 Extended Visualizations of Results in Sec. 4

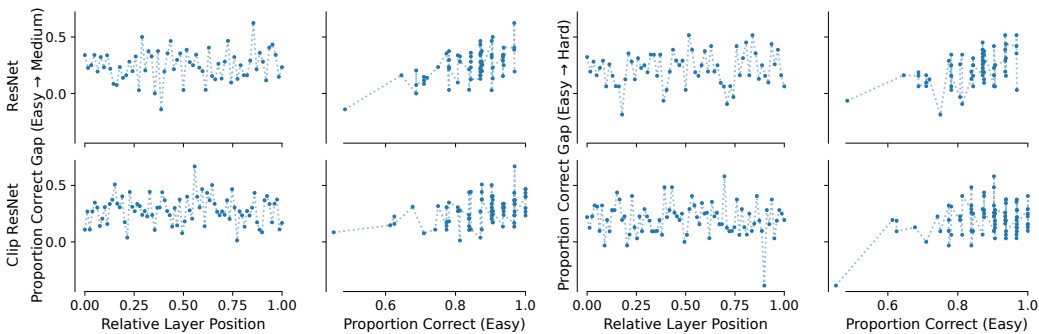

Figure 10: **Well-interpretable units do not necessarily stay interpretable in harder tasks.** For each unit investigated of the ResNet-50 (first row) and the Clip ResNet-50 (second row) model, we visualize the gap in human performance between the easy and medium (first two columns) and the easy and hard (last two columns) tasks. We show these gaps as functions of the relative layer position (first and third column) and of the human performance in the easy condition (second and fourth column).

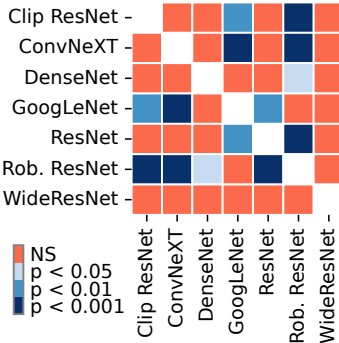

Figure 11: **Few models have significantly different interpretability scores.** The differences in interpretability afforded by synthetic feature visualizations are mostly non-significant (NS) in a Conover test with Holm correction for multiple comparisons; see Fig. 3 for significance values for natural exemplars.

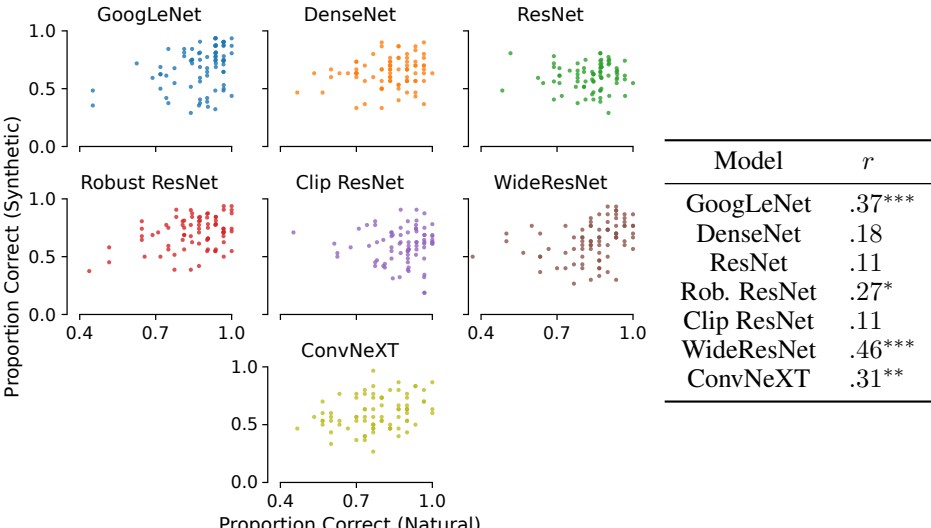

Figure 12: **Measured interpretability using different methods is partially correlated.** We investigate how the interpretability measured in our psychophysical experiment for the explanation method of natural dataset samples is predictive for that measured using synthetic feature visualizations. The table shows Spearman's rank correlation between the proportions correct when using natural and synthetic explanations. Asterisks denote significant correlations. While we see a strong correlation for some models, this does not hold for all.

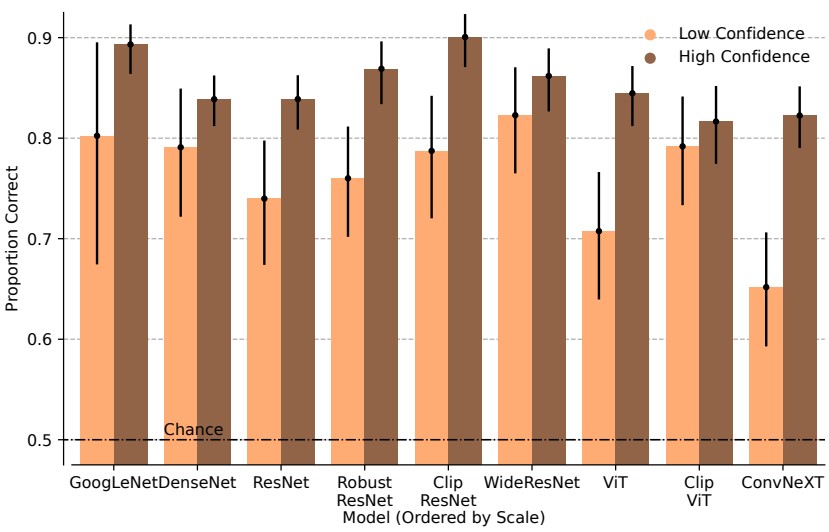

(a) Natural exemplars.

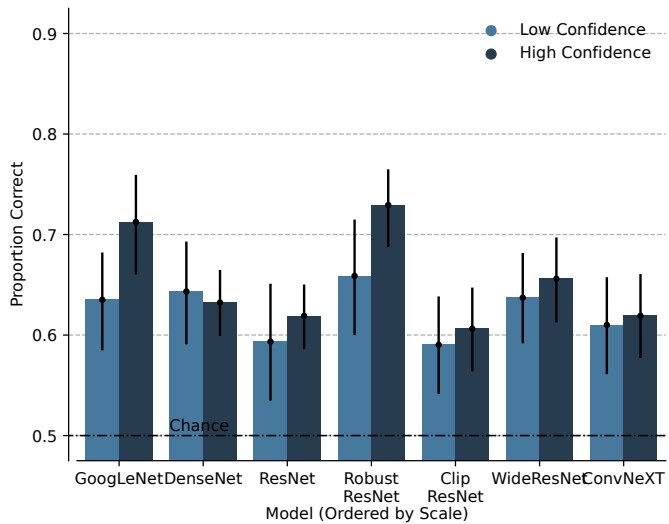

(b) Synthetic feature visualizations.

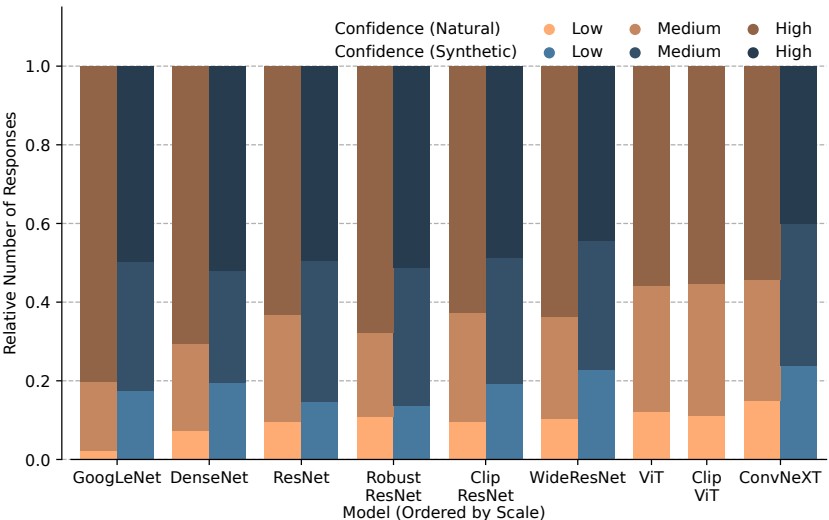

(c) Distribution of confidence ratings.

Figure 13: **More confident responses are mostly more correct.** We investigate the relationship between the confidence indicated by the participants and the correctness of the given response. For this, we compare the proportion correct for responses with low (i.e., = 1) and high (i.e., = 3) confidence ratings for all models and both natural exemplars (a) and synthetic feature visualizations (b). For the natural exemplars (a), we find that for almost all models, a higher proportion of responses are correct when the associated confidence ratings are higher. For the synthetic condition (b), this only holds for two models, if at all. Additionally, the distribution of confidence ratings (c) shows that natural examples lead to higher confidence scores for all models.

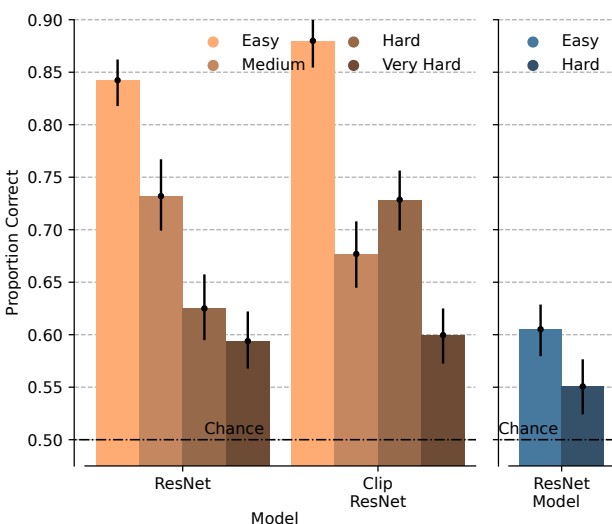

Figure 14: **Impact of unit sampling on performance.** In Fig. 6, we investigate the effect of task difficulty on performance. Due to an oversight, not all 80 units sampled for this experiment were kept identical between the difficulty levels, but only 63. Here, we visualize the result for only those units that were shared between the difficulty levels. The inconsistency has no relevant qualitative effect on the conclusion: Performance rapidly declines as the task becomes harder.

## B.2 Analysis of Quality Checks

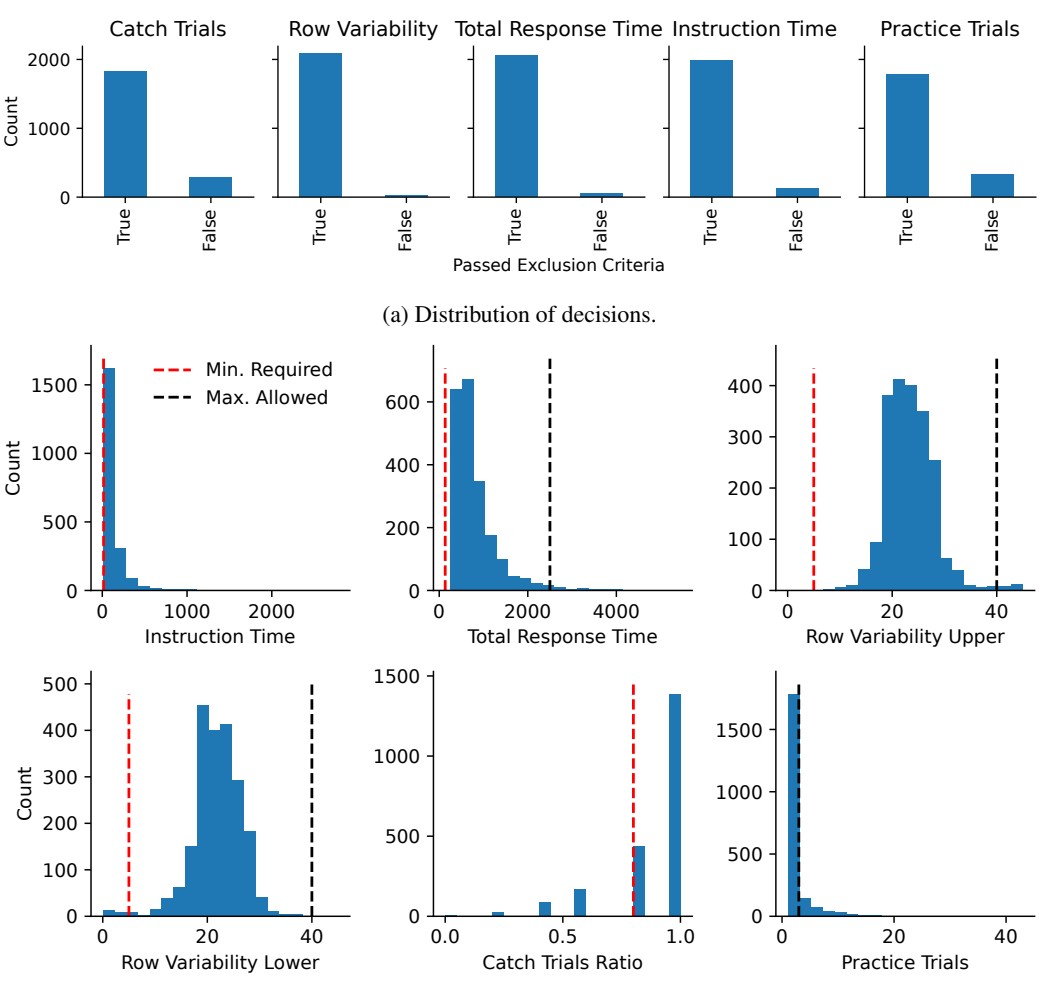

(a) Distribution of decisions.

(b) Distribution of values used for decision.

Figure 15: **Most participants pass quality checks.** For each of the five quality checks outlined in Appx. A.3, we show a distribution over the number of participants that have passed/failed this check (top) and the distribution over the values used by the checks. The black and red lines in the latter indicate the minimally required and the maximally allowed values, respectively.

## B.3 Distribution of Activations

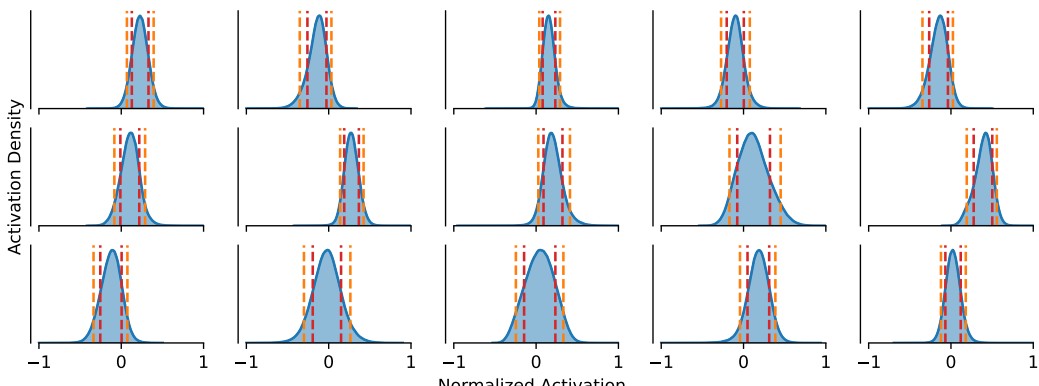

Figure 16: **Activation distribution is unimodal.** We display the distribution of activation for 15 randomly chosen units from GoogLeNet. The activations have been divided by the largest absolute activation per unit to restrict the distribution to values between $-1$ and $1$. The orange and red lines indicate the location of the 85th and 95th percentile as well as that of the 15th and 5th percentile, respectively. It is apparent that the distribution is unimodal and does not feature multiple pronounced peaks/modes at its tail.

## B.4 Are Activation Patterns in Feature Maps Predictive of a Unit's Interpretability?

Since we observe large differences in unit-wise interpretability across all networks, a logical research direction is to find out what drives these differences. As an example, we investigate two hypotheses here.

**Contrast.** First, we investigate whether there is a relationship between a unit's interpretability and the local contrast in the activation maps of convolutional layers caused by validation set images. This is motivated by the idea that if a feature is concentrated at one location in the image, it might be easier to be detected by human observers than if the activation is distributed across the image.

We visualize the relationship between a unit's interpretability and the computed contrast in its activation maps in Fig. 17. There does not appear to be a strong relationship between the two, as supported by low Spearman's rank correlations ($-0.24 \leq \rho \leq 0.14$).

**Sparseness.** Second, we analyze whether the sparseness of activations in a feature map is predictive of a unit's interpretability. This is motivated by the argument that units that sparsely fire over a large dataset are sensitive to a particular image feature that might be easier for humans to detect and understand.

To test this, we investigate two measures of sparseness: First, we compute the fraction of non-positive values (i.e., zeros after ReLU activation) in a unit's feature map averaged over the ImageNet validation set. The resulting data and the units' interpretability scores are shown in Fig. 18. As for the contrast baseline, we see only a weak, non-significant relation between the two. Second, we compute the fraction of images in the ImageNet validation set for which an entire feature map achieves only non-positive values (i.e., zeros after ReLU activation). Analogously to before, the resulting data is shown in Fig. 19, and we find no strong relationship.

## C Broader Impacts

We expect the broader impacts of our work to be positive since advancements made with respect to the interpretability of AI systems should increase their transparency and fairness. However, as is always the case for interpretability work, explanations can also give users a false sense of trust in the explained model. This can lead to the deployment of models that, under real-world conditions, give incorrect or undesired results. Too much trust in AI systems can also lead to their deployment in areas

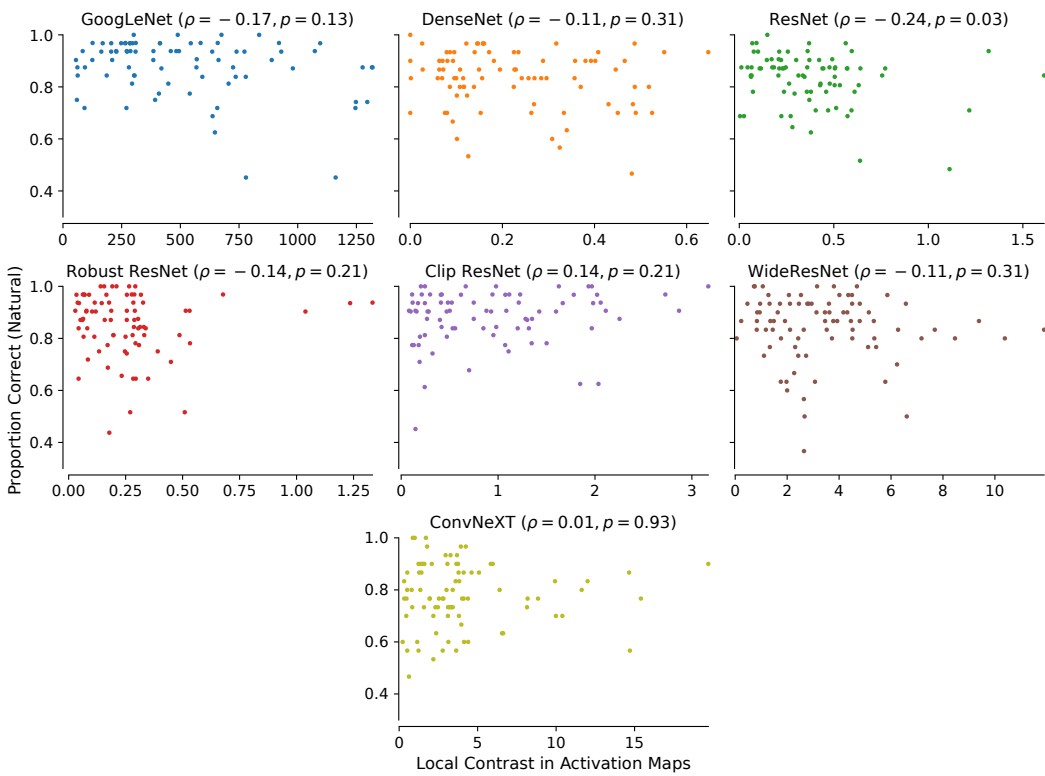

Figure 17: **Local contrast of activation maps does not predict a unit's interpretability.** We compute the average local contrast in the activation maps caused by validation set images for the sampled units of the investigated convolutional networks. The units' interpretability, measured by the proportion correct, does not appear to be a function of the local contrast.

that are better left in human hands for ethical reasons, such as policing or the justice system. Apart from these general and high-level concerns, we see no direct way in which someone could use the findings and data presented here to cause harm, especially since we do not build an interpretability method but investigate whether models are interpretable.

## D    Computational and Financial Cost

The most computationally intensive aspect of this work is creating stimuli for the experiments, which can be further subdivided into collecting natural exemplars and producing feature visualizations. The former point is negligible since all that is required is one forward pass over the ImageNet training set for each model. We record the activations on Nvidia 2080Ti GPUs and perform multiple forward passes due to memory constraints, but even if we assume a pessimistic 4 hours of GPU time and full utilization of the GPU at 250 W, this results in 9 kWh power consumption for all models in total. Creating feature visualizations for 100 randomly selected units — we later randomly sample 84 units for each model and kept some stimuli for anticipated later experiments — requires the parallel use of 25 2080Ti GPUs for about 12 hours for all models except ConvNeXt, which takes about 24 hours on average. Since this is done for only seven models because we do not generate feature visualizations for the ViTs, the required electricity amounts to 600 kWh. Assuming our country's consumer electricity price of 0.4812 € / kWh and the country's typical $CO_2$ emissions per kWh of 428 g CO2e / kWh, both of which are pessimistic estimates given that the experiments ran in a local academic datacenter, these requirements translate to about 300 USD and 256 kg of CO2 equivalent emissions.

The financial cost of this work is dominated by crowdworker compensations. As outlined in Appx. A.3, workers are compensated at an hourly wage of 15 USD, or 2.79 USD / HIT. Since all workers are

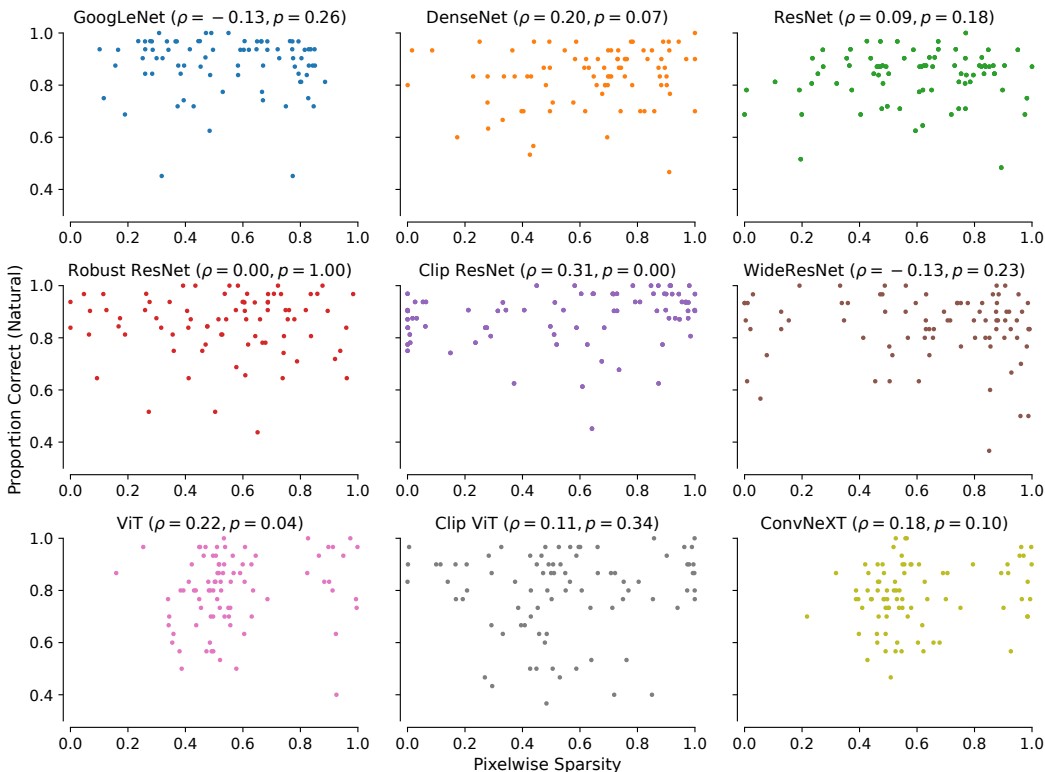

Figure 18: **Sparseness of activations does not predict a unit's interpretability.** We compute the fraction of non-positive values (i.e., zero after ReLU activation) in the feature maps of the units of interest averaged over the ImageNet validation set for all investigated models. We then show a unit's interpretability as a function of this pixel-wise sparseness measure. However, the two do not appear to have a meaningful relationship, as indicated by Spearman's rank correlation shown above each plot.

compensated, even if the results of their HIT do not pass our quality checks, the total cost incurred by the experiment (including the fees paid to MTurk) amounts to around $12'000$ USD.

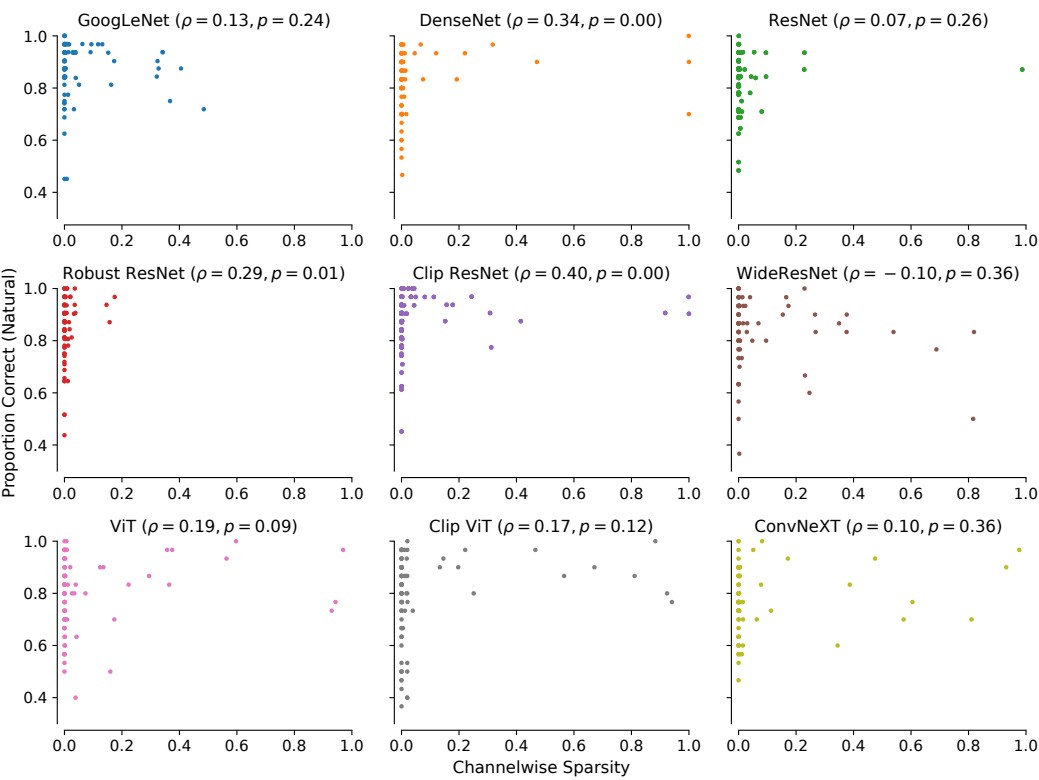

Figure 19: **Sparseness of entire channels does not predict a unit's interpretability.** Similar to Fig. 18, we compute the fraction of images for which an entire feature map achieves only non-positive values (i.e., zero after ReLU activation). Analogously to before, we plot a unit's interpretability as a function of the channel-wise sparseness and find no strong relation between this sparseness measure and a unit's interpretability.

# E Further Screenshots of Psychophysics Trials

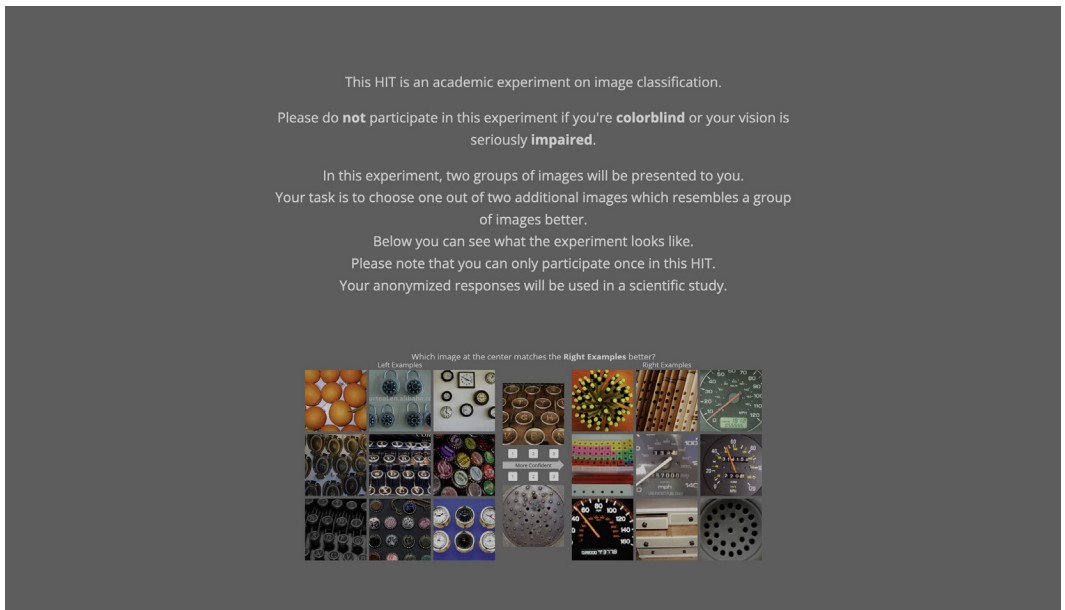

Figure 20: Screenshot of the initial overview of the HIT presented to workers considering the task. We inform participants that they consent to their anonymized data being used for a scientific study.

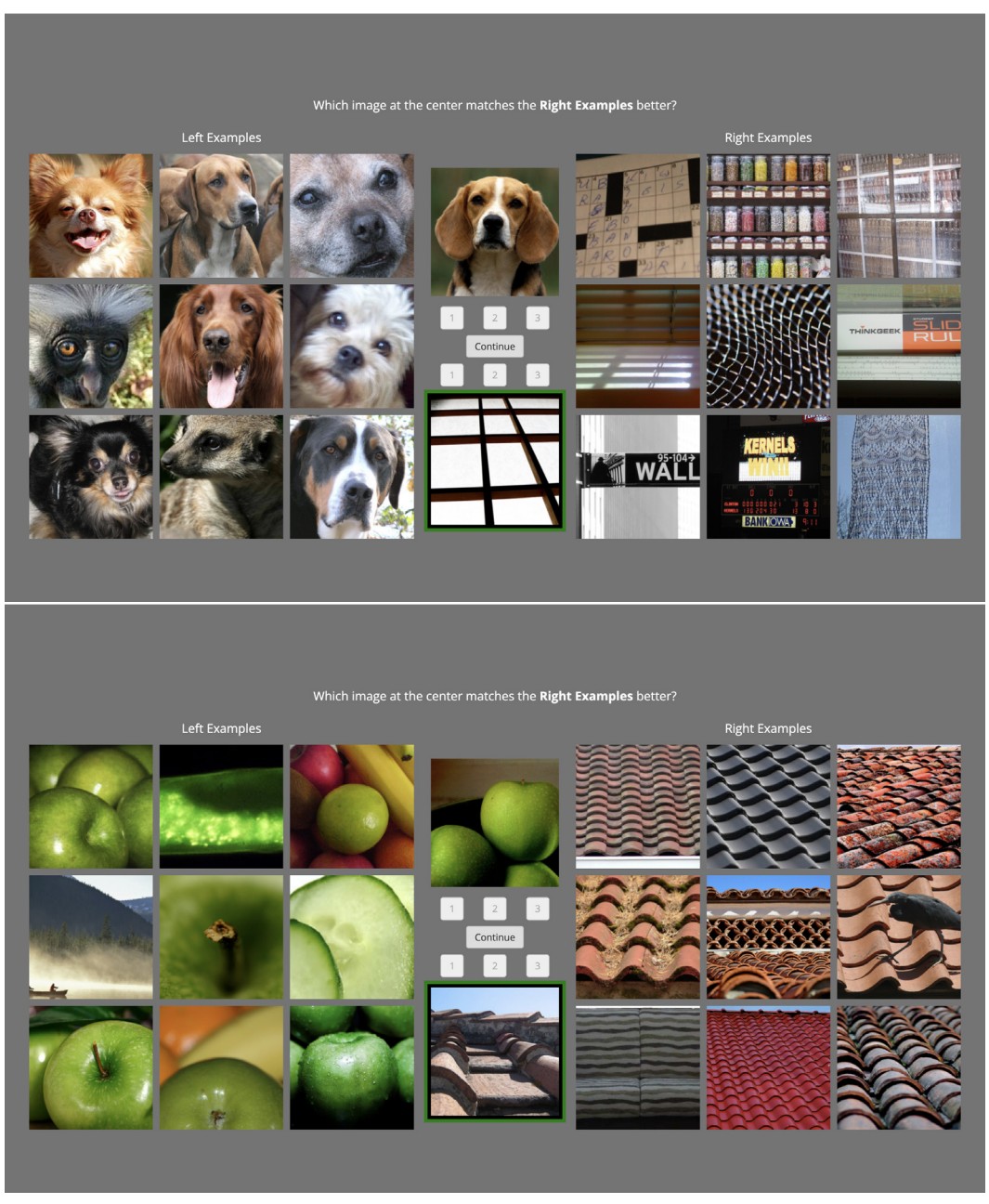

Figure 21: Screenshots of two of the twelve possible instruction trials to explain the task to participants in the natural condition after the participant has given the correct response. See Fig. 22 for examples in the other condition.

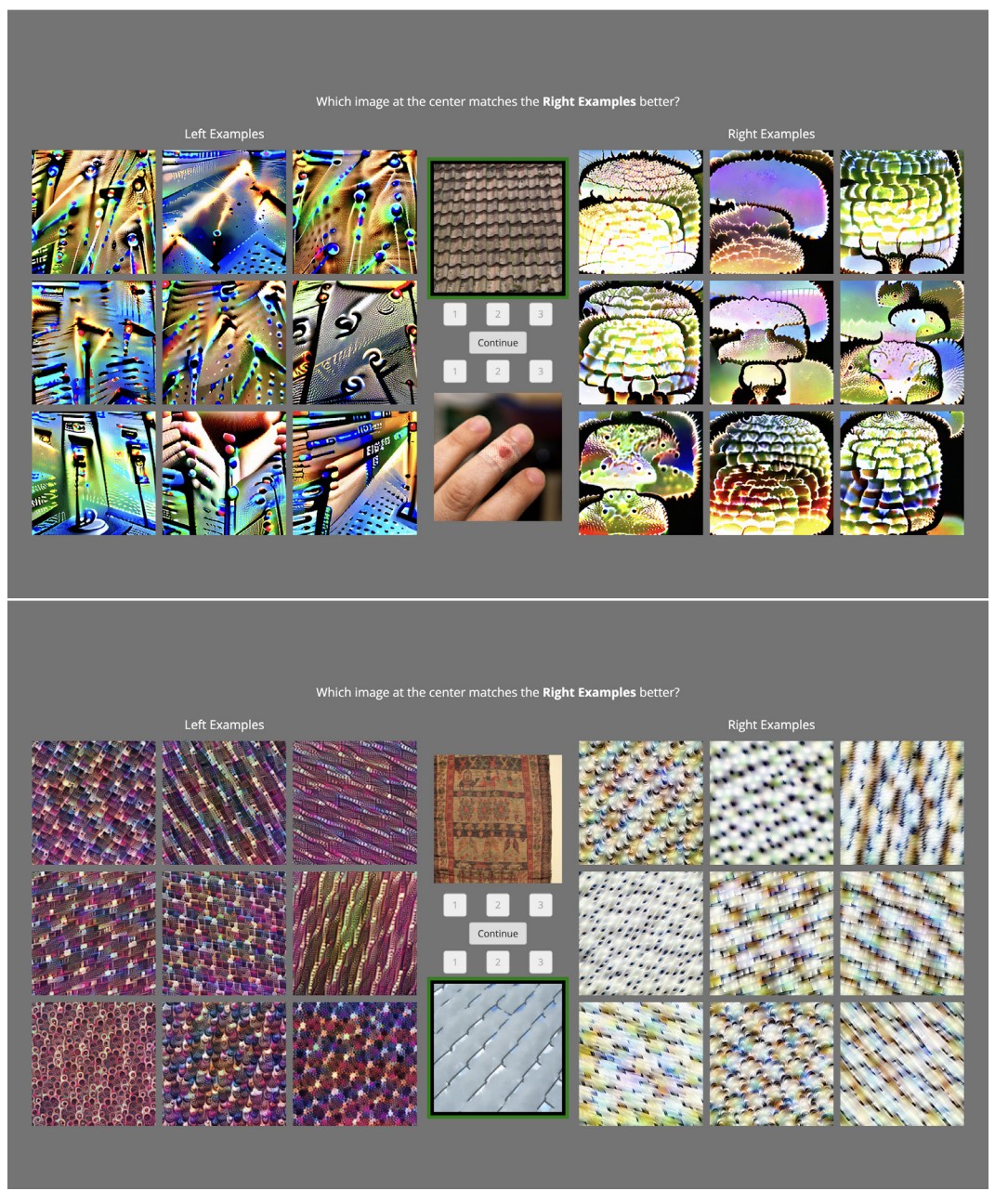

Figure 22: Screenshots of two of the twelve possible instruction trials to explain the task to participants in the synthetic condition after the participant has given the correct response. See Fig. 21 for examples in the other condition.

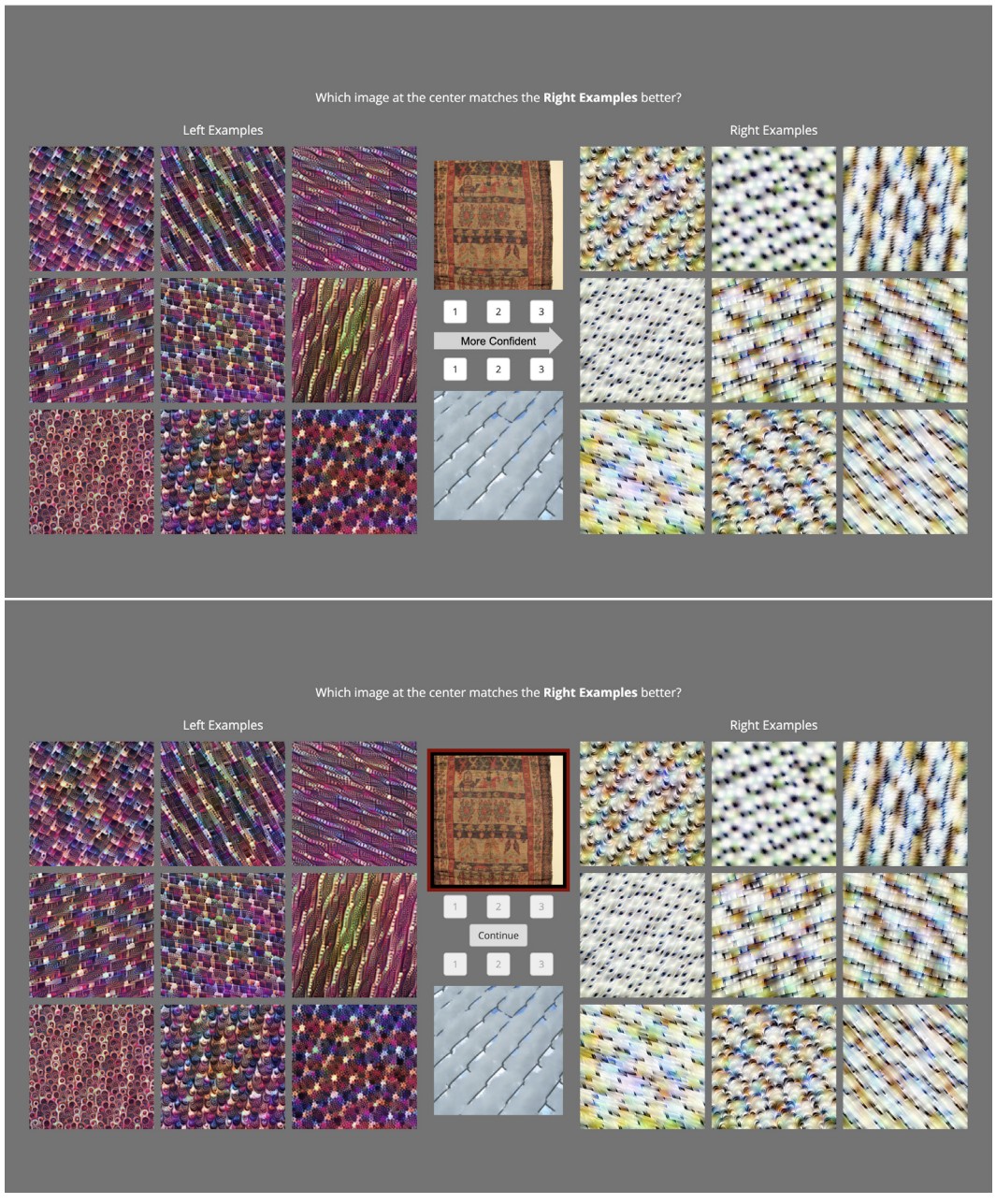

Figure 23: Screenshots of two of the twelve possible instruction trials to explain the task to participants in the synthetic condition before the participant has given a response (top) and after the participant has given the wrong response (bottom).

