# OpenReview forum: "Scale Alone Does not Improve Mechanistic Interpretability in Vision Models"
_NeurIPS.cc/2023/Conference — NeurIPS 2023 spotlight_

### Official Review · Reviewer_6B8X · 2023-06-23

**Soundness:** 4 excellent
**Presentation:** 3 good
**Contribution:** 3 good
**Rating:** 7
**Confidence:** 4

**Summary:**

This paper presents an empirical evaluation of how interpretable the individual neurons in different models are, with specific focus on how increasing model scale effects this. This is done via large scale human experiments via amazon mechanical turk, and they find no evidence that larger models would be more interpretable, instead having weak evidence towards large models being less interpretable. They also release their evaluation data to help create automatic evaluation in the future via models to predict interpretability to a human.

**Strengths:**

- Sound experimental setup and statistical analysis
- Studying an important problem
- Larger scale than previous experiments, finding additional evidence towards previous findings such as feature visualizations being less informative than natural images, and robust models having more informative feature visualizations
- Clearly written
- Good discussion of related work

**Weaknesses:**

I think the experimental setup could be improved in a few ways:
- I think 84 neurons per model, while bigger than previous might not be enough, especially if wanting to compare between layers in a single model, as this leaves less than 2 neurons per layer for ResNet-50 for example, which is not large enough sample size. This might also be part of the reason why most findings lack statistical significance.
- While the costs make it harder to scale, could have been better to narrow down focus i.e. look at fewer models/layers within these models or use fewer evaluators per neuron.
- Top-5% and top-15% activations for medium and large are too big of a jump in my opinion, top-15% rarely has any meaningful pattern for a neuron. Would be more interesting to see closer results like in the top-1% range

Finally results are not surprising, I wasn't expecting scale alone or the other features studied here to have a significant effect on the interpretability. This reduces the impact of the work but it is still good information to have confirmed.

**Questions:**

- How many trials were conducted? The main paper states more than 100k, but appendix A.3 only states 40k, where is the difference?

- Why was the alternative image chosen from least activating images? Why not for example use a randomly chosen image as the other option? Or perturbed versions of highly activating images as done by Zimmermann et al.?

**Limitations:**

Yes, good discussion.

---

> ### Author Rebuttal · Authors · 2023-08-08
>
> Dear Reviewer,
>
> Thank you for your constructive review of our work and for describing it as *“studying an important problem”* with a *“sound experimental setup/analysis”*.
>
> Please find our responses to your points below:
>
> **Q:** *“How many trials were conducted”* \
> **A:** We apologize for the confusion: The numbers in Appendix A.3 corresponded to a preliminary version of our dataset and were outdated. We updated this section now: “In total and excluding pilot experiments, we collect data for 120'745 trials, of which 69'330 pass the quality checks.”.
>
> **Q:** *“Why was the alternative image chosen from the least activating images”* \
> **A:** Thank you for raising this question. We followed the reasoning of [1], who propose to use both strongly and weakly activating images as query images with the goal of making the task easier for humans. When the human participants are presented with both of these query images, there are effectively two valid strategies for solving the task: either by seeing that the weakly activating image is indeed weakly activating or by seeing that the strongly activating query image is strongly activating. By not showing the weakly activating query image, we could make the task harder - however, as the human performance is not yet reaching the ceiling performance of 100%, we argue that the current task difficulty is chosen reasonably.
>
> **Q:** *“84 units per model might not be enough”* \
> **A:** We also pondered this question and performed an a priori power analysis (Sec. A.5) to decide on how many units to test while balancing the cost of the overall experiment. We based our analysis on what we believed to be a relevant effect size (a difference in group means of 10 percentage points) so that significance would also imply relevance. We could have sampled more units per model at the expense of dropping a few models from the comparison, but all that would have meant is that smaller differences would have become statistically significant. We updated our manuscript to include a discussion of this more prominently.
>
> **Q:** *“Top-5% and top-15% activations [...] are too big of a jump [...]. Would be more interesting to see closer results like in the top-1%”* \
> **A:** Thank you for this insightful comment. We agree that the top-15% is a rather big jump both in activations and, expectedly, also in task difficulty. This setting was meant to test the limit of the usefulness of existing per-unit interpretability methods; hence, we also included the more conservative setting of top-5%. Nevertheless, we now conducted another experiment for the top-1% and updated Figure 7 (see Figure 7 of the attached PDF). Here, we also see a substantial (albeit smaller than for the top-5%) drop in performance for both models investigated. We updated our manuscript accordingly.
>
> **Q:** *“not surprising [...] but good information to have [this result] confirmed”* \
> **A:** We are happy that you find our results interesting! Whether or not the finding is perceived as surprising, of course, depends: We believe there are reasonable grounds to hypothesize that scaling up models and datasets promotes more human-aligned feature extraction/processing given that the overall model behaviors also get more aligned with humans, according to [2,3]. Testing this hypothesis was the primary motivation for this work. We integrated this reasoning more clearly in the main text.
>
> [1]  Exemplary Natural Images Explain CNN Activations Better than State-of-the-Art Feature Visualization. Borowski et al. ICLR 2021. \
> [2] Partial success in closing the gap between human and machine vision. Geirhos et al. NeurIPS 2021. \
> [3] Scaling Vision Transformers to 22 Billion Parameters. Dehghani et al. ICML 2023.

---

> > ### Comment · Reviewer_6B8X · 2023-08-12
> >
> > Thanks for the response!
> >
> > This has mostly addressed my concerns and I am happy to see results with random samples from top-1% images, pretty much as expected but good to see and gives more nuance to the results. I will be keeping my score as it is as it reflects my view that this is a solid paper that should be accepted but not necessarily an outstanding one.

---

### Official Review · Reviewer_q3wq · 2023-07-03

**Soundness:** 2 fair
**Presentation:** 3 good
**Contribution:** 2 fair
**Rating:** 7
**Confidence:** 3

**Summary:**

In this paper, the author performed a large-scale psychophysical experiment to see whether scales improve mechanistic interpretability. They found that there is no scaling effect for interpretability; the latest large models do not provide better mechanistic interpretability than older models. The dataset with user response is publicized so that it can be used in future studies that aim to optimize interpretability.

I think the paper is novel in that it answers a new question that was not explored in this way. However, since it lacks new theoretical results, substantial methods, or findings, I believe the NeurIPS datasets and benchmarks track would be a more suitable venue for the paper.

**Strengths:**

1. The approach to answering the question of whether scales affect mechanistic interpretability via large-scale psychological experiments is novel.
2. The experiment was performed on various axes such as model sizes, dataset sizes, and model architectures.
3. The large-scale psychological experiment dataset would be a valuable resource for further research
4. analyzed the dataset using multiple ways.
5. The paper was well written.

**Weaknesses:**

1. The paper does not provide substantially new methods or findings.

**Questions:**

1. Is the hypothesis that scaling improves mechanistic interpretability a common perception?

**Limitations:**

1. I am not sure whether the hypothesis that scaling improves mechanistic interpretability is a common perception. If not, the utility of the dataset would be limited.

---

> ### Author Rebuttal · Authors · 2023-08-08
>
> Dear reviewer,
>
> Thank you for your helpful review and for acknowledging the novelty and breadth of our experimental work as well as for describing our dataset as a *“valuable resource for further research”*.
>
> Please find our responses to your points below:
>
> **Q:** *“Is the hypothesis that scaling improves mechanistic interpretability a common perception?”* \
> **A:** This is a valid question, and we understand your concern about the prevalence of this hypothesis in the research community. We have updated our manuscript to better convey our motivation to investigate this question. Previous work [1, 2] demonstrates that an increased scale results in models that show more human-like behavior, as measured by error consistency. It is conceivable that this behavioral similarity is caused by more human-like feature extraction/processing strategies. However, our experimental results indicate that this is not the case: Scale does not implicitly optimize interpretability, and more human-like behavior does not automatically correspond to a more interpretable feature extraction process. We believe this insight will be of interest to a substantial part of the research community, as is also evident by the positive sentiment of the other reviewers. We updated our manuscript accordingly and included this information in the main text as we agree that this motivation is important for readers.
>
> **Q:** *“[...] does not provide substantially new methods or findings”* \
> **A:** It is true that we do not develop a new method. Instead, we see the novelty of our study in obtaining a better understanding of how different model design choices — such as model/training dataset scale and training objective — influence a model’s interpretability, as measured by a common technique. To meet this end, we scale up an established experimental paradigm and conduct a large-scale evaluation of various models and conditions. This allows us to answer the research question posed above: None of the investigated design choices actually have the effect of making models more interpretable. We believe this to be an important result, as it strongly implies that better interpretability is orthogonal to other optimization criteria and should be an explicit goal on its own.
>
>
> [1] Partial success in closing the gap between human and machine vision. Geirhos et al.. NeurIPS 2021. \
> [2] Scaling Vision Transformers to 22 Billion Parameters. Dehghani et al. ICML 2023.

---

> > ### Comment · Reviewer_q3wq · 2023-08-14
> >
> > Thanks for the response. The clarification on the motivation addressed my concern about the motivation of this work, and I've decided to increase my score.

---

### Official Review · Reviewer_fVHW · 2023-07-06

**Soundness:** 3 good
**Presentation:** 4 excellent
**Contribution:** 3 good
**Rating:** 7
**Confidence:** 4

**Summary:**

 This paper starts with the motivation of  a recent growing field of mechanistic interpretability, in which the goal is to attempt to reverse engineering a given neural network (or possibly any function). The main hypothesis is that since recent models have grown in size in terms of number of samples and number of parameters, if models perform more human-like task, the extracted features might become more interpretable. The authors then perform a large-scale human psycho/meta-physical study, in which they ask participants which of the two presented group of images are more aligned with a selected neuron’s response. The correctness of the response then serves as a metric for how interpretable a neural network is, in which they find neural networks from recent years (2017-2023), despite being larger in parameters and datasets size, are not mechanistically interpretable, which means study subjects are unable to identify the correct group of images. Last but not least, the authors release the data collected from participants as a dataset, to motivate the need for an automated formula/system to promote interpretability within any trained model.

**Strengths:**

Overall, the paper is well-written and its contents are well-organized. I did not run into trouble understanding the paper.

Interestingly, the work is heavily influenced by Olah 2017, which looks into activation maps of individual neurons of a neural network and find “object detectors” such as “dog neurons.” It is also somewhat parallel to works done back in 1950s, when neuroscientists would probe neurons in monkey visual cortex and find “edge” detectors from simple/complex cells. Hence I believe this experiment, although somewhat similar to what previous has done in terms of methodology, is an important contribution to the field of mechanistic interpretability, and to the entire field of interpretability/study of neural networks on a unit level in general. Although the connection to neuroscience in mentioned in the related work, personally I would suggest adding a short sentence to the introduction regarding its parallel to neuroscience.

The setup of the experiment is also reasonable and sound. Authors have sufficiently considered the potential biases of the experiment and have addressed them accordingly in the design of the experiment. The experiment is also done with a wide range of models (both CNNs and ViTs) and datasets of different scales. I believe the experiments are well thought out and done.


**Weaknesses:**

In this work, the authors constructed the hypothesis “If models make more human-like decisions, this might hint at a closer alignment between the extracted features and human perception.” While ImageNet classification is often considered standard for benchmarking, it is hard to argue that better models of predictive performance/larger models implies more human-like decisions. A better benchmark would have been comparing models that can perform multiple tasks, which humans do. An alternative solution would be to perform the same experiment on architectures, but that would obviously induce a much higher financial cost. I think the authors can better address why ImageNet is an indicative task for human-like decision in their Method or Introduction.

Small typo. In Training Details on OpenReview website, authors wrote: “we do not train but onyl evaluate models”, onyl -> only

But other than the above, there are no major weakness in this work.


**Questions:**

1. Have the authors considered performing a similar experiment but with language tasks, since language is a more natural form of interpretability and method of showing one’s understanding? For instance, finding nearest neighbors of activation maps (as vectors), and ask whether the neighbors form any semantic clusters.
2. The definition of mechanistic interpretability stated by the authors: “understanding the internal information processing of deep neural networks by attempting to reverse engineer them.” It is not immediately clear to me what it means when the authors stated in the conclusion that we need to optimize for this property. Can the authors elaborate on that? Does it imply we should design neural networks differently? Does it mean we should change the learning objectives we use to train the networks? Do the authors mean that we should design models with the ability to reverse engineer them? But wouldn’t that imply we already know how neural network works?


**Limitations:**

The authors have adequately addressed the limitations in their Limitation section.

---

> ### Author Rebuttal · Authors · 2023-08-08
>
> Dear Reviewer,
>
> Thank you for your constructive review and for assessing our work as a *“well-written”* and *“important contribution to the field of interpretability”* with *“reasonable and sound”* experiments.
>
> Please find our responses to your points below:
>
> **Q:** *“I would suggest adding a [...] sentence to the introduction regarding its parallel to neuroscience”* \
> **A:** You are right that the field of mechanistic interpretability has been inspired by approaches from neuroscience. We edited our manuscript and added this information in the introduction.
>
> **Q:** *“Have the authors considered [...] language tasks”* \
> **A:** Thank you for the suggestion. We did not investigate language in this work but agree that it would be one of the most natural and dense formats for an explanation. We have updated the Related Work section to include a paper [3] that takes steps towards using natural language to summarize sets of highly activating exemplars.
>
> **Q:** *“It is not immediately clear [...] what it means [...] that we need to optimize for this property [interpretability]”* \
> **A:** Thank you for raising this point. You are correct that we mean one needs to consider model interpretability throughout the model design and training and cannot simply hope to obtain an interpretable model through existing/commonly used techniques for scaling models up. Here, we use the term “interpretable models” to refer to models where one has knowledge about what each unit in the network does and how it contributes to the overall feature extraction process for a substantial part of the network. You are also correct that this knowledge would drastically simplify reverse engineering networks. We extended the conclusion of our manuscript to include this information and explain the point more clearly.
>
>
> **Q:** *“Why is ImageNet an indicative task of human-like decisions?”* \
> **A:** This is an excellent question, and we have edited our manuscript to clarify our stance on this matter, as we did not mean to imply that ImageNet performance is indicative of human-like decision-making. Having said this, there is previous work [1,2] suggesting that scaling up models and datasets results in models that exhibit more human-like behavior, as measured by error consistency. We argue that it is conceivable that this behavioral similarity is caused by more human-like decision strategies, which in turn are characterized by their reliance on (non-spurious) features. Hence, we are interested in investigating whether this also results in a more interpretable feature extraction process. To make this connection more direct, we now integrate a new plot that shows the per-unit interpretability of a model as a function of their “human-likeness” measured by an established benchmark [1] (see Figure 4 of the attached PDF). We have updated our manuscript accordingly and included this information in the main text.
>
> [1] Partial success in closing the gap between human and machine vision. Geirhos et al. NeurIPS 2021. \
> [2] Scaling Vision Transformers to 22 Billion Parameters. Dehghani et al. ICML 2023.

---

> > ### Comment · Reviewer_fVHW · 2023-08-17
> >
> > Thank you for your response and I have no further questions.

---

### Official Review · Reviewer_fNyx · 2023-07-06

**Soundness:** 3 good
**Presentation:** 4 excellent
**Contribution:** 4 excellent
**Rating:** 7
**Confidence:** 5

**Summary:**

The authors investigate whether features associated with neurons in vision models become more or less interpretable as these models are scaled up. They utilize mechanical turk and a forced association test to see whether human workers are able to clearly associate exemplar images (or synthetic feature activations) to strongly vs. weakly activating sets of input images for given neurons. They find that across multiple scales and classes of vision models, there is no significant increase in interpretability when increasing scale. They provide a number of additional findings on the shortcomings of feature visualizations, and trends (or lack thereof) in interpretability as a function of neuron specificity and the location of neurons within a network. In addition to these interesting findings, they release an annotated dataset of neurons and their associated intepretabilities which should prove a great boon to mechanistic interpretability researchers more generally.

**Strengths:**

Overall this paper provides a meaningful contribution to the field of interpretability research; both through its own investigations and through the accompanying release of the IMI Dataset.

## Originality and Significance
The psychoanalytic approach toward addressing questions about interpretability has thus far been relatively underexplored, and is used to great effect in this work. The findings themselves seem general enough to guide future research on interpretability methods for vision models, but the release of the associated dataset may be more impactful still, as it will allow researchers with fewer resources to work on superior algorithmic approaches for discovering features / Interpretable neurons.

The additional experiments carried out in App. B.4 are also very interesting, as these undermine some intuitions prevalent in the community regarding which kinds of simple flags are likely to indicate interpretable neurons (though the annotations don’t explicitly measure the levels of polysemanticity, thus allowing this to remain as a potential confounder).

## Quality and Clarity
The methodology and findings are mostly well-presented, and only minor changes are suggested (see Weaknesses)


**Weaknesses:**

The methodological approach seems comprehensive and well-suited towards addressing the questions of interest. I have no major criticisms.

The two minor criticisms are on:
* **Claims** - the paper repeatedly makes claims of the form “mechanistic interpretability of larger models is not better”, which is supposed to be short-hand for “the interpretability of features associated with neurons in larger vision models is not better”. I think this amounts to an overstatement of the valid claim without being any more concise, as reframing “feature interpretability” to “mechanistic interpretability” is a generalization (there are other forms of mechanistic interpretability which may be unaffected by these findings, such as circuit discovery). Collapsing “larger vision models” to “larger models” is fine, given that this is obvious. Additionally, one might wish to argue that “feature interpretability” is too broad, but the field’s current definition of features is quite loose, and the paper acknowledges issues around superposition and synthetic feature generation, so this is fine.

* **Presentation** - The paper is very well presented for the most part. The only part that was slightly difficult to parse was Fig.2 - here I found the examples in the appendices more illuminating (partly due to a nitpick regarding the caption, mentioned below). I don’t think there would be any loss of generality if this figure were replaced with one of these examples from the appendix (as the caption explicitly states that the query may be a natural exemplar or synthetic visualization)

## Nitpicks:
* 74: -> “was their usefulness .. experimentally quantified”
* 114: do you mean “higher density of interpretable units”?
* Fig 2 Caption: It was slightly unclear at first that “two extremely activating” meant one extremely positively, and one extremely negatively activating; this made the following “pick the more positively activating query” slightly confusing.
* Fig 3 Caption: The caption is split into part “A” and “B”, but these are not labelled pictographically. Changing to “left” and “right” would be easiest.
* 147 - “keep” -> “kept”, as latter part of sentence is past tense
* 174 - “paradigm shift” seems quite strong
* 300 - The footnote referencing superposition should be places before the period


**Questions:**

It would be interesting to know how the effectiveness of synthetic feature visualizations varies for different models, as there is some discussion of the difficulties involved in using existing methods to generate features for the different models. Figure 12 mostly answers this as one can infer the “difference between synthetic and natural interpretability for each model” but it is not directly shown. It might be nice to add a figure with e.g. “3 non-cherry picked examples of synthetic feature visualizations” for each model, alongside the difference in synthetic vs natural interpretability. Perhaps this is not interesting as the feature visualizations are generally quite poor (as per Sec. 4.2).

**Limitations:**

Limitations are thoroughly discussed.

---

> ### Author Rebuttal · Authors · 2023-08-08
>
> Dear reviewer,
>
> Thank you for your thorough review and your thoughtful comments. We were delighted by your enthusiasm for our IMI dataset and by your assessment of our work as a *“well-presented”*  and *“meaningful”* study with *“very interesting findings”*.
> We also updated the caption of Figure 2 to present the setup more clearly and incorporated your other suggestions.
>
> Please find our responses to your points below:
>
> **Q:** *“reframing ‘feature interpretability’ to ‘mechanistic interpretability’ is a generalization”* \
> **A:** Thank you for raising this point. We agree that using precise expressions is important, and we did not mean to overstate the generality of our work. We used ‘mechanistic interpretability’ as a short form for understanding how neural networks work. As there are different approaches for this, we now rephrased our paper to more precisely characterize our paradigm as a sub-form of mechanistic interpretability (i.e., ”per-unit mechanistic interpretability”) to avoid overly general statements.
>
> **Q:** *“How [does] the effectiveness of synthetic feature visualizations var[y] for different models?”* \
> **A:** Thank you for this suggestion. We included a new figure demonstrating typical feature visualizations by randomly choosing three units per model and depicting a positive and a negative feature visualization for them in Appendix A5 (see Figure 10 of the attached PDF for a preview). We find no obvious and substantial degradation of the feature visualizations for any model.

---

> > ### Comment · Reviewer_fNyx · 2023-08-14
> >
> > Dear Authors,
> >
> > Thank you for your response. I consider all questions satisfactorily addressed and will maintain my rating which confidently recommends this paper as a meaningful contribution to the field of interpretability research.
> >
> > The additional supplemental figure is interesting and certainly answers my question clearly!

---

### Author Rebuttal · Authors · 2023-08-08

We would like to thank all reviewers for their time and their constructive feedback. We appreciate their assessment of our work as *“an important contribution to the field of mechanistic interpretability”* with *“reasonable and sound experiments”* (fVHW), a *“great boon to mechanistic interpretability researchers”* with *“interesting findings”* (fNyx) and a *“novel approach”* producing *“valuable resource[s]”* (q3wq). All reviewers seemed to agree that our work was *“well/clearly written”* (q3wq, fVHW, 6B8X), *“well organized”* (fVHW), and *“well-presented”* (fNyx). We hope that our work is indeed *“general enough to guide future research on interpretability methods for vision models”* (fNyx).

Inspired by your helpful comments, we incorporated the following changes into the next/final revision of our paper:

- We conducted an additional experiment to better capture the decrease in subject performance as the task difficulty increases, sampling query images from the 99th percentile (see Figure 7 of the attached PDF).
- We added a figure to the Appendix that offers a qualitative impression of the feature visualizations used in the experiments (see Figure 10 of the attached PDF).
- We updated the caption of Figure 2 to increase the clarity of this illustration and to make the visualization of our task setup more accessible.
- We created a new figure that analyzes how predictive human-aligned behavior (measured by [1]) is for high per-unit interpretability scores (see Figure 4 of the attached PDF).
- We updated our manuscript to fix minor mistakes and reduce the risk of misunderstandings.

---

### Decision · Program_Chairs · 2023-09-21

**Decision:**

Accept (spotlight)

**Comment:**

This paper studies vision models that differ in scale, architecture, training paradigm and dataset size, and wants to know whether those factors influence the mechanistic interpretability of those models as evaluated in a large scale study on mechanical turk. They find that none of these factors has an effect on the interpretability of individual units in the setting they evaluated. In other words, more recent, better performing models are not automatically more interpretable in this setting. They release their dataset and call for models to be designed to be more interpretable. The reviewers were unanimously positive about this paper, and mentioned it was well written, interesting to interpretability researchers, has well designed experiments, a good set of vision models, and could guide future research in (interpretable) vision models.